# Single-cell profiling reveals a memory B cell-like subtype of follicular lymphoma with increased transformation risk

Xuehai Wang[1,8], Michael Nissen[1,8], Deanne Gracias[1], Manabu Kusakabe[1], Guillermo Simkin[1], Aixiang Jiang[2], Gerben Duns[2], Clementine Sarkozy[2,3], Laura Hilton[2], Elizabeth A. Chavez[2], Gabriela C. Segat[1], Rachel Wong[1], Jubin Kim[1], Tomohiro Aoki[2], Rashedul Islam[4], Christina May[4], Stacy Hung[2], Kate Tyshchenko[1], Ryan R. Brinkman[1], Martin Hirst[4], Aly Karsan[4], Ciara Freeman[2], Laurie H. Sehn[2], Ryan D. Morin[4,5], Andrew J. Roth[6], Kerry J. Savage[2], Jeffrey W. Craig[2], Sohrab P. Shah[6,7], Christian Steidl[2], David W. Scott[2] & Andrew P. Weng[1] ✉

Follicular lymphoma (FL) is an indolent cancer of mature B-cells but with ongoing risk of transformation to more aggressive histology over time. Recurrent mutations associated with transformation have been identified; however, prognostic features that can be discerned at diagnosis could be clinically useful. We present here comprehensive profiling of both tumor and immune compartments in 155 diagnostic FL biopsies at single-cell resolution by mass cytometry. This revealed a diversity of phenotypes but included two recurrent patterns, one which closely resembles germinal center B-cells (GCB) and another which appears more related to memory B-cells (MB). GCB-type tumors are enriched for *EZH2*, *TNFRSF14*, and *MEF2B* mutations, while MB-type tumors contain increased follicular helper T-cells. MB-type and intratumoral phenotypic diversity are independently associated with increased risk of transformation, supporting biological relevance of these features. Notably, a reduced 26-marker panel retains sufficient information to allow phenotypic profiling of future cohorts by conventional flow cytometry.

Follicular lymphoma (FL) is one of the most common types of indolent non-Hodgkin lymphoma. Management ranges from watchful waiting to rituximab-based combination therapy for symptomatic or threatening disease. Regardless of the therapeutic approach applied, in addition to expected relapse or progression of FL disease, there is an inherent risk of transformation to a more aggressive B-cell lymphoma, most commonly diffuse large B cell lymphoma (DLBCL), at a rate of 2–3% per year[1]. Disease progression or transformation is thought to occur as a result of evolution of pre-existing clones, or emergence of new clones that have managed to evade, nullify, or co-opt the host immune response[2]. Baseline clinical features such as FLIPI score, performance status, and B symptoms are informative for risk of transformation;[3,4] however, identifying biological predictors at diagnosis has proven elusive[2].

[1]Terry Fox Laboratory, BC Cancer Agency, Vancouver, BC, Canada. [2]Lymphoid Cancer Research, BC Cancer Agency, Vancouver, BC, Canada. [3]Drug Development Unit, Institut Gustave Roussy, Villejuif, France. [4]Canada's Michael Smith Genome Sciences Centre, BC Cancer Agency, Vancouver, BC, Canada. [5]Department of Molecular Biology and Biochemistry, Simon Fraser University, Burnaby, BC, Canada. [6]Molecular Oncology, BC Cancer Agency, Vancouver, BC, Canada. [7]Epidemiology and Biostatistics, Memorial Sloan Kettering Cancer Center, New York, NY, USA. [8]These authors contributed equally: Xuehai Wang, Michael Nissen. ✉e-mail: aweng@bccrc.ca

We describe here a phenotypic analysis of both malignant B-cell and infiltrating T-cell populations from patient lymph node (LN) biopsies involved by FL using mass cytometry (CyTOF)[5]. The relatively large size of our study cohort which includes 155 diagnostic FL plus 36 normal, or reactive LN (rLN) specimens enabled discovery of variable but also recurrent phenotypes among patient samples. Integrated analysis with clinical outcome information reveals features associated with risk of transformation, thus supporting the utility of highly dimensional single-cell phenotypic profiling.

## Results

### Patient samples

From 2013 through 2017, we identified a total of 155 patients with FL for whom cryopreserved cells were available from their initial, pre-treatment diagnostic excisional biopsy specimens. All but 6 were LN specimens (Supplementary Data 1). Patient characteristics for the identified sample cohort are shown in Supplementary Table 1. We accessed an additional 36 rLN biopsies with age/sex/anatomic site

distribution comparable to the 155 FL cohort and deemed non-malignant on pathologic review to serve as normal controls (Table S1 and Supplementary Data 1).

### Global multi-dimensional analysis readily segregates normal and malignant B-cell populations

After data pre-processing (Supplementary Fig. 1), global mapping of all 191 samples (155 FL + 36 rLN) in UMAP[6] space revealed that normal and malignant B-cells occupied largely distinct regions of phenotypic space (Fig. 1A). There was remarkably limited phenotypic variation across the 36 rLN samples which is highlighted by their high Shannon entropy (Fig. 1B). Cells from FL samples on the other hand generally occupied areas high, intermediate, and low entropy. High entropy areas co-localized with rLN cells (dashed red lines in Fig. 1B), suggesting these represented residual normal B-cells in FL samples. Intermediate entropy areas (solid red lines in Fig. 1B) suggested two abnormal, but recurrent, phenotypes adopted by FL cells from different patients. The remainder of cells occurred in areas of low

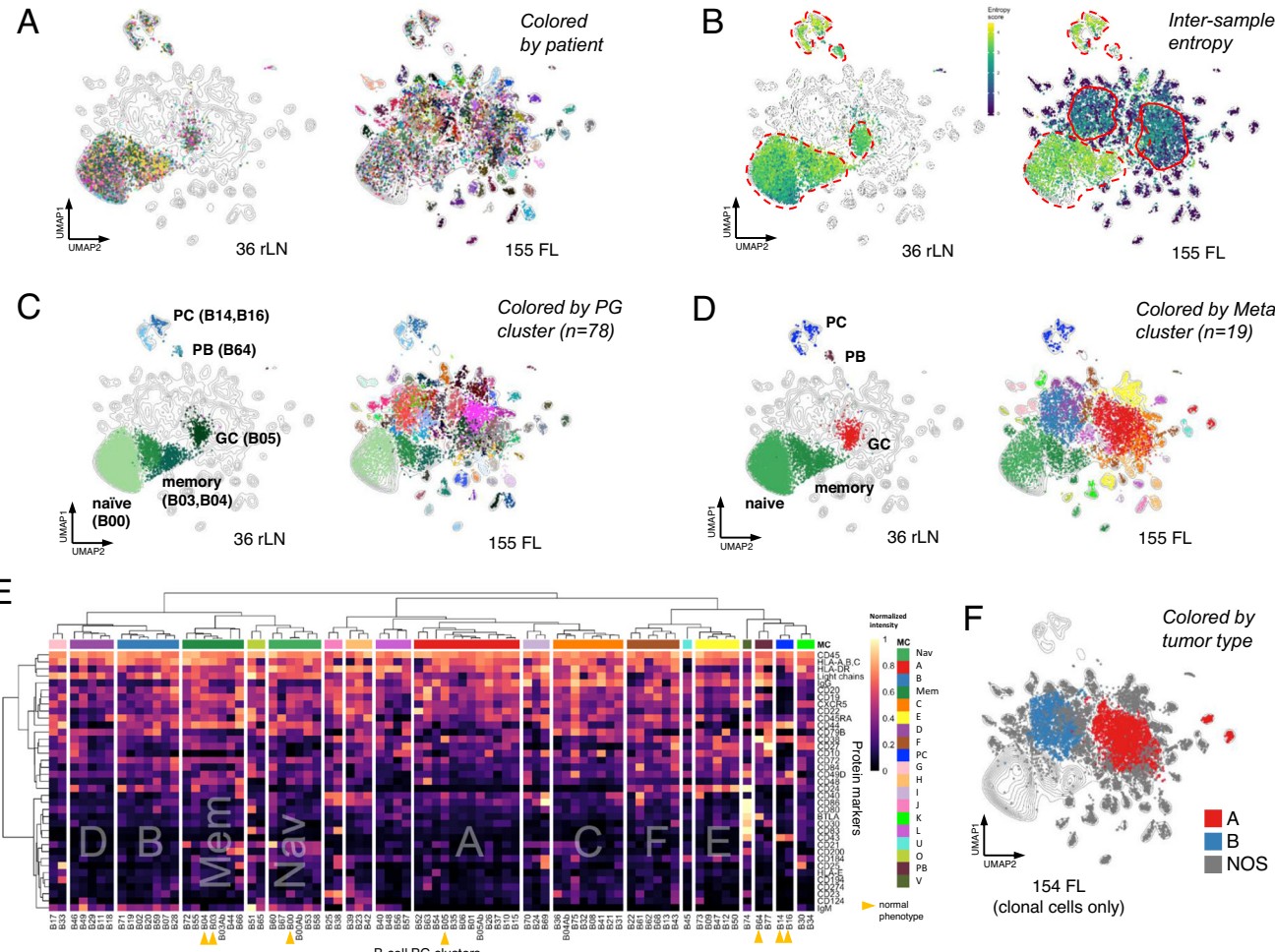

**Fig. 1 | Highly dimensional single-cell phenotyping reveals both variable and recurrent phenotypes in FL. A** Dimensional reduction UMAP plots of B-cell data from combined analysis of 36 rLN + 155 FL patient samples. Each dot represents a single cell. Dots are colored by patient sample. Equal numbers of cells from rLN and FL groups were randomly sampled for display. Contour lines show density in the combined dataset. **B** UMAP plots as in (**A**) but with cells colored by Shannon index to reveal mixing of cells from different patients (intertumoral entropy). Broken red lines highlight regions co-occupied by rLN cells. Solid red lines highlight two regions of high entropy unique to FL samples. **C** UMAP plots as in **A** but with cells colored according to their assigned PhenoGraph (PG) cluster. Normal B-cell populations in rLN samples were manually annotated based on expression of

reference marker proteins. **D** UMAP plots as in **A** but with cells colored by their assigned MetaCluster (MC). **E** Heatmap of relative protein expression by CyTOF from all rLN and FL samples. Each row is a protein marker, each column is a B-cell PG cluster. B-cell clusters are hierarchically clustered into 19 MetaCluster (MC) groups. MC-Mem and MC-Nav include normal memory (B03, B04) and naïve (B00) B-cells; MC-A includes normal GC B-cells (B05). **F** UMAP plot as in **A** but with cells colored according to one of 3 defined tumor types. Individual samples with >50% clonal B-cells assigned to MC-A, MC-B, or neither were designated as types A, B, or NOS. FL follicular lymphoma, rLN reactive lymph node, GC germinal center, PB plasmablast, PC plasma cell, PG Phenograph, MC metacluster, NOS not otherwise specified, UMAP Uniform Manifold Approximation and Projection.

entropy, each often containing cells from just a single patient (Fig. 1A, B).

We next sought to cluster the cells in a manner that recapitulated the two dominant areas of tumor cells with moderate entropy as noted in the UMAP projection. To accomplish this, we first applied PhenoGraph (PG)[7] to define phenotypically distinct clusters among B-cells across the full dataset which yielded 78 separate PG clusters (Fig. 1C and Supplementary Fig 2). While this level of clustering segregated individual populations of tumor cells with low entropy from one another well, the two dominant areas of moderate entropy appeared to be over-clustered. We thus applied hierarchical clustering of the 78 PG-level clusters to obtain 19 meta-clusters, or MCs (Fig. 1D, E and Supplementary Figs. 3 and 4), where 19 represented the optimal number of clusters as measured by the gap statistic[8]. This defined two dominant MCs which comprised 25% and 15% of all B-cells, respectively, and corresponded roughly to the two areas of moderate entropy seen in Fig. 1B. We designated these two dominant MCs as MC-A and MC-B (colored red and steel blue, respectively, in Fig. 1D).

Cells from rLN samples segregated into 7 different PG clusters which mapped into 5 different MCs and could be annotated as naïve, memory, and germinal center (GC) B-cells, plasmablasts, and plasma cells based on canonical markers of normal B-cell differentiation[9–13] (Supplementary Fig. 5a). There were two distinct PG clusters within the memory B-cell meta-cluster which could be annotated as pre- and post-switch memory B-cells based on expression of IgG in the latter (Fig. 1E). The majority of rLN samples exhibited a relatively consistent balance of naïve, memory, and GC B-cells (Supplementary Fig. 6). FL samples, in contrast, were often devoid of normal GC B-cells and showed increased memory B-cells as compared to rLN, revealing that accumulating malignant B-cells distort the normal balance across B-cell compartments.

Among the 155 FL samples, about one-quarter of cells (23%) were assigned into the same 7 PG clusters as cells from rLN samples (Fig. S2 and Supplementary Data 2), suggesting they could represent residual, normal B-cells within the tumor-involved lymph nodes. To verify if these cells were indeed normal, we examined their surface light chain expression pattern, defining B-cell polyclonality by kappa:lambda ratios between 7 and 0.3 (Fig. S7a)[14]. Based on this definition, the majority exhibited polytypic light chain expression (356 sample-level clusters across 155 FL); however, a subset exhibited kappa:lambda ratios >7 or <0.3 (79 sample-level clusters across 155 FL), including many which mapped as GC B-cells. These monotypic versions of otherwise phenotypically normal B-cell clusters were designated with an Ab(normal) suffix (e.g. B05Ab). Of note, the monotypic light chain expressed by Ab clusters consistently matched the monotypic light chain of phenotypically aberrant clusters in the same sample (Supplementary Fig. 7b). One interpretation of these Ab cells is that they may be part of, or alternatively a precursor to, the established malignant clone; however, the alternate possibility that they represent non-malignant, transient monotypic expansions that ultimately self-resolve cannot be excluded. One FL sample did not contain any detectable monotypic B-cells by these criteria, thus reducing the number of informative FL samples to 154 in total.

## Unsupervised clustering identifies two recurrent subtypes of abnormal B-cells in FL

The initial unsupervised PG clustering defined two clusters, B01 and B02, that contained cells from 75 (49%) and 34 (22%) of 154 informative FL samples, respectively, revealing these two phenotypes are particularly common and shared across many different FL patients. This finding stands in striking contrast with our recent study of DLBCL where each patient's tumor is essentially unique when mapped in 39-dimensional phenotypic space[15]. PG clusters B01 and B02 mapped into metaclusters MC-A and MC-B, respectively (Fig. 1E). MC-A also

subsumed normal GC B-cells (PG cluster B05) which would be compatible with the conventional notion of FL as being closely related to GC B-cells. MC-B, on the other hand, did not subsume any normal B-cell clusters and was clearly distinct from GC B-cells, exhibiting phenotypic features closer to pre-class switch recombination (CSR), IgM+ IgG- memory B-cells (Fig. 1E and Supplementary Data 3).

The two most populated MC groups, MC-A and MC-B together comprised half of all malignant B-cells (31% and 19%, respectively). Phenotypic positions of less populated MC types C-F tended to emanate outward from the more centrally located MC types A and B (Supplementary Fig. 4), while MC types Mem and Nav were located in close proximity to, and in fact subsumed their corresponding normal memory and naïve B-cell PG clusters, respectively (Supplementary Figs. 4 and 1E). The top markers discriminating between MC-A and MC-B, and these two from all other cell types included elements of the B-cell receptor (BCR; IgM, IgG, KL, CD79B), major histocompatibility complex (MHC) complex (HLA-DR), and signaling/signaling modulators CD44[16], CD24[17], and CD22[18] (Supplementary Fig 5b and Supplementary Data 4). For instance, cells of type MC-A tended to express IgG, HLA-DR, and CD22, whereas those of type MC-B tended to express IgM/CD79B/KL and CD24/CD44. Markers discriminating the remaining MC types are also provided in Supplementary Data 4 and sometimes include BCR, MHC, and CD44/CD24/CD22 markers, depending on their proximity to MC-A and MC-B.

It is worthy to note that while MC-A and MC-B describe phenotypes of cells seen across FL samples, any given patient tumor sample may be composed of a mixture of cell types. As a convention, we assigned tumor types according to their most abundant MC cell-type component, by which 136/154 tumors (88%) were composed of at least 80% cells of the assigned MC type (Supplementary Fig. 8). The lowest MC-A and B contents of tumors assigned to types A and B were 67.5% and 64% with the next most abundant MC types being MC-L (32.5%) and MC-I (33.2%), respectively (Supplementary Data 5). When defined in this manner, 28% of FL samples (43/154) would be considered type A and 18% (28/154) as type B. Tumors of the remaining MC types were considerably less abundant with 15, 12, 11, 10, 5, and 4 samples assigned to types C, D, E, F, Mem, and Nav, respectively (Supplementary Data 5). There was no discernible commonality to these less abundant tumor MC types and we designated them as type NOS (not otherwise specified) in order to focus subsequent analyses on distinguishing features of the more abundant types A and B (Fig. 1F). As expected, tumors lumped together into the NOS category were phenotypically heterogeneous, in contrast to the relative homogeneity seen within types A and B (Fig. S9).

## Orthogonal validation of FL subtypes by single-cell RNA-Seq

Given the unexpected finding of two highly recurrent, distinct subtypes of FL, we sought to validate this distinction by an independent approach not limited by our particular selection of 39 CyTOF markers. We thus performed single-cell (sc) RNA-Seq on 4 rLN and 6 FL samples, the latter of which were selected from the CyTOF cohort to include relatively pure examples of types A and B (Fig. 2A; see Fig. S10 for a summary of all available data types for each sample). B-cells from rLN and FL samples again mapped to largely distinct areas from one another, with the exception of limited numbers of residual normal B-cells in FL samples, while T-cells from rLN and FL samples were largely co-incident. PhenoGraph identified 18 clusters, 12 of which could be annotated as normal B- or T-cell subsets[19] (Fig. 2B and Supplementary Fig 11). As in CyTOF data, the 4 included type A FL cases all mapped in very close proximity with one another and partially overlapped with normal GC B-cells from rLN samples. The 2 included type B FL cases mapped separately from the type A cases and showed closer proximity to non-GC B-cell subsets as measured by Pearson correlation-based distance (Fig. 2C). Supervised analysis to identify RNAs that most discriminated between abnormal B-cells from type A

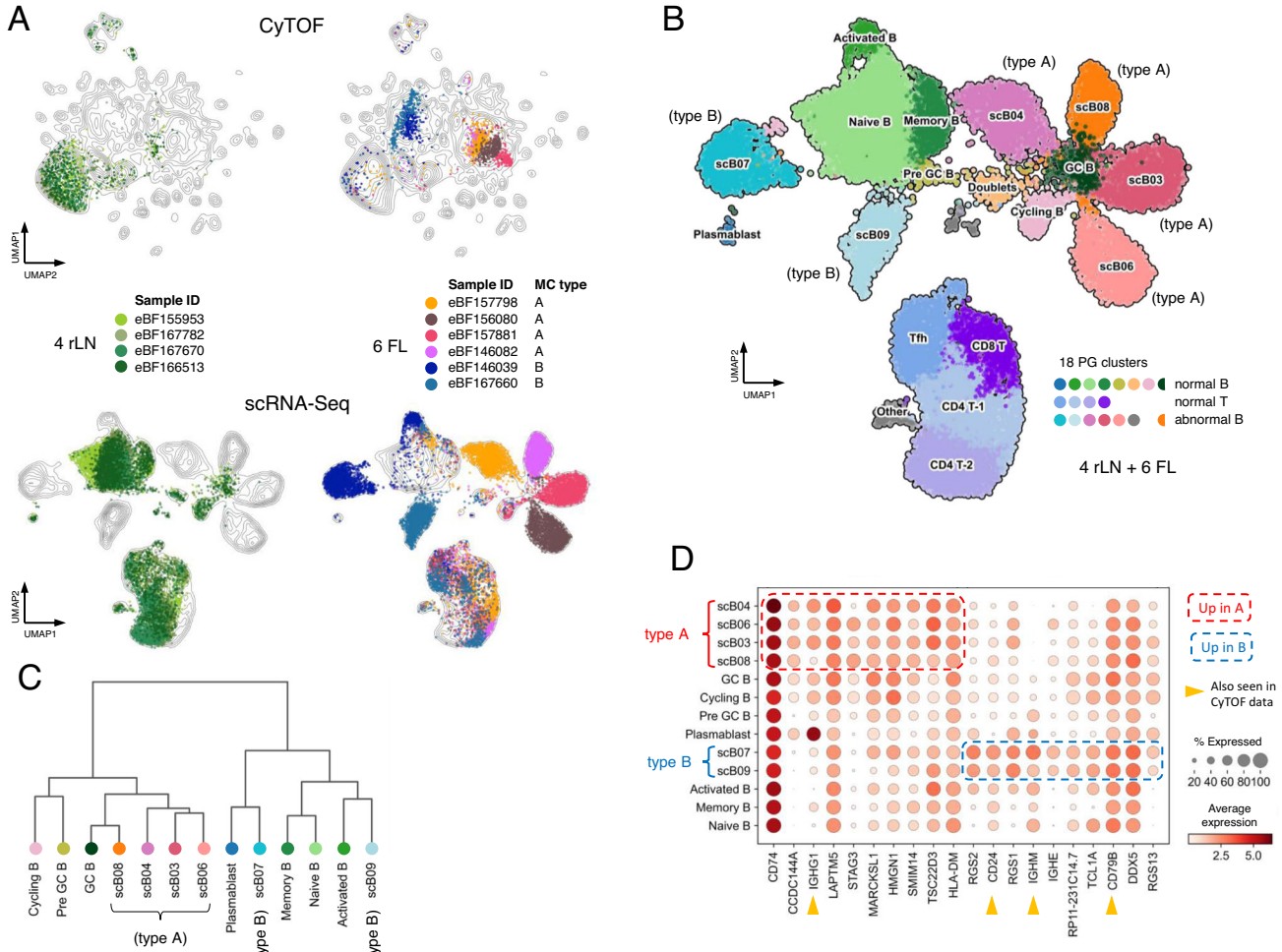

**Fig. 2 | Single-cell RNA-seq recapitulates CyTOF-defined FL subtypes. A** Parallel analysis by CyTOF and scRNA-seq. Upper panels show UMAP plots of CyTOF data as in Fig. 1A. Lower panels show UMAP plots of scRNA-seq data for the same set of samples. Each dot represents a single cell. Dots are colored by patient sample. Contour lines show density in the combined rLN + FL datasets. The indicated MC types were defined by CyTOF. **B** UMAP plot of scRNA-seq data as in lower panels of **A** but merging all 10 samples into the same plot. Dots are colored by their assignment into one of 18 scRNA-seq clusters. Normal B- and T-cell clusters were manually annotated based on expression of reference marker RNAs. scB clusters were unique to FL samples. Cells in the GC B cluster were assigned to the same cluster as scB08 but derive from rLN samples and are labeled separately for clarity.

**C** Dendrogram of Pearson correlation distances between singlet B-cell scRNA-seq clusters using Ward's method. Colors correspond to cell populations as indicated in **B**. **D** Mean expression levels of genes differentially expressed between abnormal B-cells from FL type A (scB03, scB04, scB06, scB08) vs type B (scB07, scB09) samples. Top 10 genes expressed in each of FL type A and B cells are depicted. Normal B-cell subsets are also included for reference. RNAs corresponding to proteins assessed in the CyTOF B-cell panel are indicated by yellow arrowheads. FL follicular lymphoma, rLN reactive lymph node, PG Phenograph, MC metacluster, GC germinal center, sc single cell, UMAP Uniform Manifold Approximation and Projection.

and B tumors confirmed some of the informative protein markers from CyTOF including IgM, IgG, CD79B, and CD24 (Fig. 2D). These results support that type A and B tumors represent distinct subtypes of FL as assessed in unbiased whole transcriptomic space and render unlikely the possibility that they represent an artifact unique to the particular selection of markers used for CyTOF analysis.

The scRNA-Seq data also presented the opportunity to explore what underlying biological differences may exist between type A and B FL cells. We thus performed differential gene expression analysis comparing abnormal B-cell populations from the 4 type A vs. 2 type B FL samples (Supplementary Fig. 12 and Supplementary Data 6). Reactome pathway analysis highlighted enrichment of translation-related ribosomal protein genes and phagocytic immune response genes in type A cells, while antigen presentation and heat shock/stress response genes were enriched in type B cells (Supplementary Data 6). Along with inspection of the component genes from these pathways, these findings suggest a basic difference could be that type A and B cells may correspond to late and early phases of the GC reaction, respectively.

We also performed a similar analysis using bulk RNA-seq data from whole tissue or unfractionated cell suspension material from type A and B samples which highlighted extracellular matrix remodeling genes in type A samples and chemokine signaling in type B samples (Supplementary Fig. 13 and Supplementary Data 6), suggesting that local microenvironmental interactions likely also differ between the two FL types. Functional studies will be needed to explore these possibilities further.

## Sample-level analysis reveals recurrent patterns of tumor cell phenotypes

Identification of common B-cell phenotypes shared across different FL samples provided the opportunity to address whether there might also be recurrent patterns of cellular composition across different tumors. To pursue this question, MCs occupied by at least 1% of malignant B-cells in each tumor were tabulated and frequencies of co-occupancy for each MC pair were calculated across all tumors and compared to their expected pairwise probability distributions[20]. MC groups that

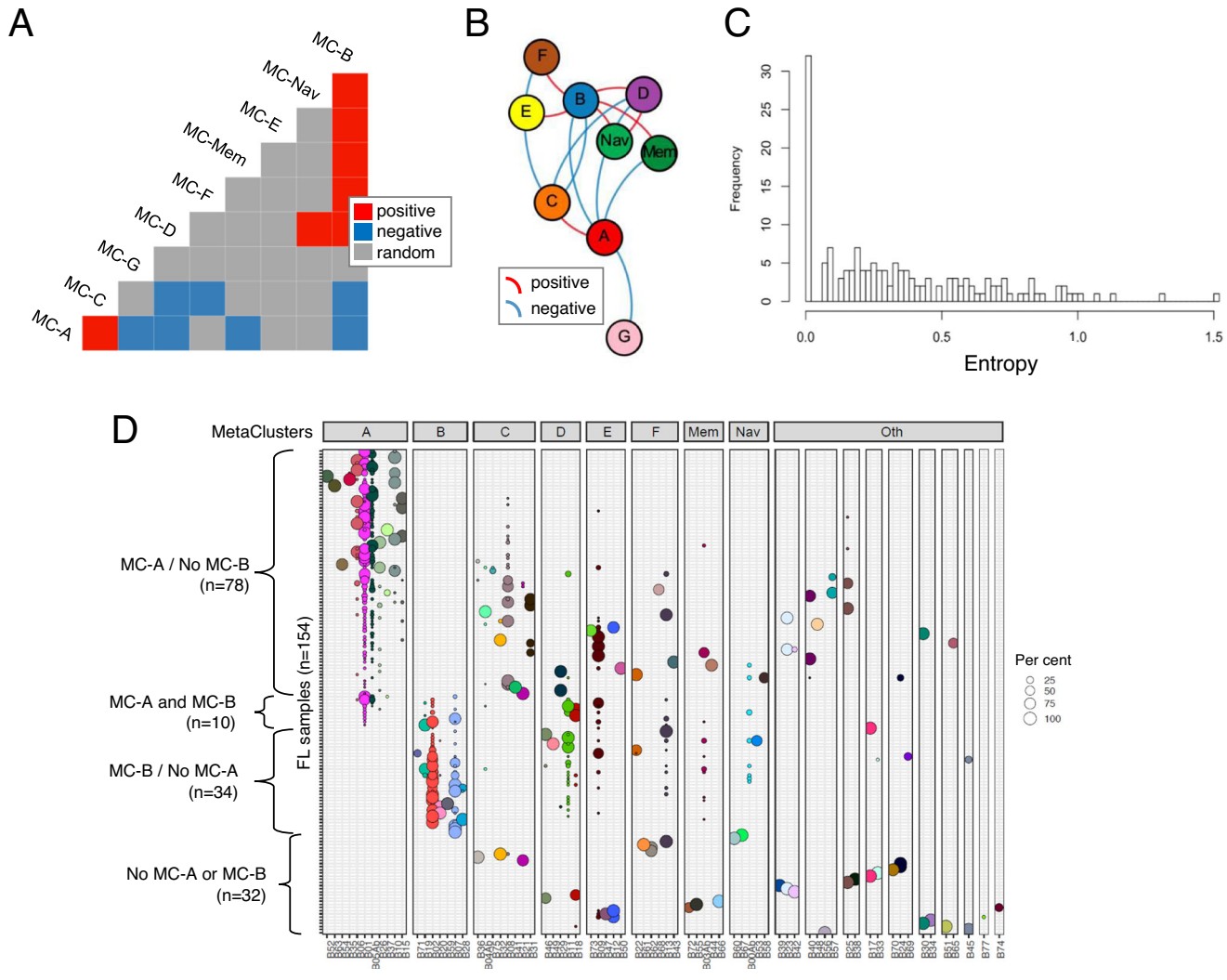

**Fig. 3 | B-cell cluster composition analysis reveals recurrent patterns within samples. A**, **B** Co-occurrence analysis for B-cell MC groups. Probabilistic model-based co-occurrence analysis was performed using malignant B-cells only from 154 FL samples without additional adjustment for multiple testing (see Methods section). **A** Co-occurrence matrix. Blue and yellow shaded boxes indicate statistically significant positive and negative co-occurrences, respectively ($p < 0.05$; probabilistic model-based co-occurrence test, 2-sided). **B** Co-occurrence network diagram. Force-directed graphing of co-occurrence associations. Each node represents a B-cell MC group. Edges indicate statistically significant positive or negative co-occurrence associations ($p < 0.05$; probabilistic model-based co-occurrence test, 2-sided). Lengths of edges and positioning of nodes are determined by a combination of attractive and repulsive forces which in sum are proportional to the strength of pairwise associations. **C** Intratumoral entropy score distribution among all FL samples ($n = 154$). Tumors with low entropy score contain cells mapping to a single PG, while tumors with high entropy scores contain cells mapping to up to 8 different PGs. **D** Composition of FL samples according to B-cell PG clusters. Bubbles within each row (= sample) indicate cells assigned to a given PG cluster (=column) with size proportional to their relative abundance in the sample (each row adding up to 100%). Bubbles are colored by PG cluster. PG clusters are grouped into metacluster (MC) groups. Samples are grouped by presence of MC-A and/or MC-B cells. FL follicular lymphoma, PG Phenograph, MC metacluster.

were co-occupied more or less frequently than expected by chance were identified and plotted in a force-directed graph (Fig. 3A, B). Co-associating MCs were often localized proximal to one another in phenotypic space (e.g. MC-A with MC-C, MC-B with MC-D/F; Supplementary Fig 4 and Supplementary Data 3). Notably, the extent of phenotypic variation was not consistent across tumors with some containing cells occupying only a single PG cluster while others occupied as many as 8 different PG clusters. We quantified this variation, taking into account the proportion of cells in different PG clusters, as intratumoral entropy (Fig. 3C). It should be acknowledged that boundaries between phenotypically adjacent MC groups may not be completely robust as by definition they dichotomize features that otherwise may potentially show continuous variation; however, on average tumors with cells exhibiting greater phenotypic variation will yield higher entropy values.

Co-occurrence analysis also identified negative correlations, most notably between the two most populated MC types A and B (Fig. 3A, B). In fact, despite MC-A and MC-B cells being present in 88/154 and 44/154 tumors, respectively (Supplementary Data 5), they co-occurred in significantly fewer samples then expected by chance alone ($\chi^2 = 19.2$, DF = 1, $p = 1.2e-5$) (Fig. 3D). This tendency for cells from MC types A and B not to co-occur within the same tumor would support the notion that they represent distinct, non-overlapping phenotypes and between which cells do not freely interconvert. Further studies will be required however to determine what if any ontogenic relation may exist between them.

### Characterization of infiltrating T-cell populations

CyTOF phenotyping was also performed in parallel on 73 of 155 FL and 34 of 36 rLN samples using a panel of 39 T-cell markers. PG clustering

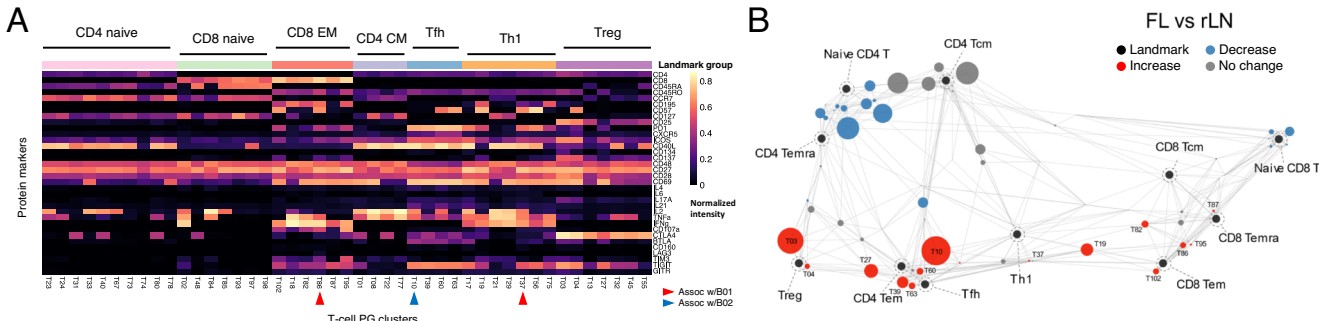

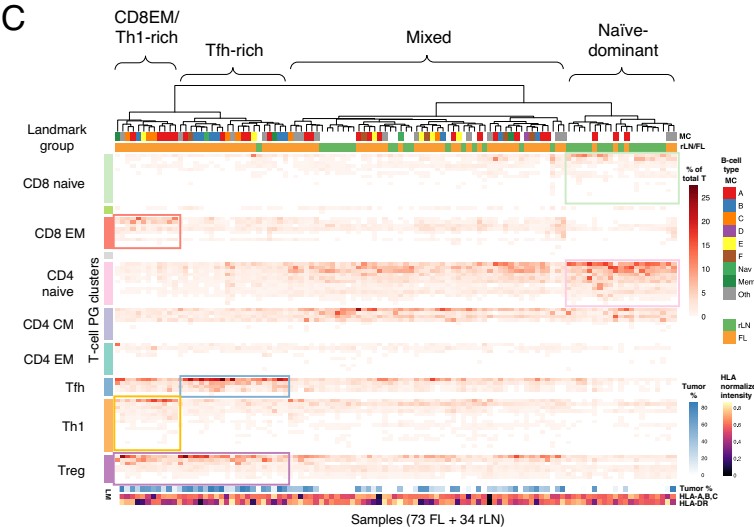

**Fig. 4 | FL samples exhibit a spectrum of T-cell states. A** Protein expression heatmap of T-cell PG clusters. Median expression values from CyTOF data are depicted for each cluster. Landmark group annotations were assigned manually by proximity in Scaffold map (**B**). The top 45 most populated PG clusters are depicted. T-cell clusters associated with B01/B02 B-cell clusters are highlighted (see Fig. 5a). **B** Scaffold map of T-cell populations in rLN and FL samples. Each node represents a T-cell PG cluster with size proportional to its median percentage across all samples. Landmark nodes are sizeless reference points defined by canonical markers. Edges are of length inversely proportional to phenotypic similarity. Nodes are colored according to statistically significant differences in abundance for the comparison FL vs. rLN (FDR < 0.05; 2-sided SAM test). **C** Sample composition by T-cell clusters. Heatmap indicates cell abundances in each sample (each column adding up to 100%). PG clusters are arranged by landmark groups. Samples are hierarchically clustered into T-cell signature groups using Ward's method on Jensen-Shannon divergence. Tumor cell % and HLA expression tracks refer to clonal B-cells within each sample. FL follicular lymphoma, rLN reactive lymph node, PG phenograph, MC metacluster, EM effector memory, CM central memory, Tem T effector memory, Temra T effector memory re-expressing CD45RA, Tcm T central memory, Tfh T follicular helper, Th1 T helper type 1, Treg, T regulatory.

yielded 85 clusters (Fig. 4A) which were mapped using a force-directed approach together with 11 canonical T-cell subsets (Fig. 4B and Supplementary Fig. 14). As compared to rLN, FL samples were most notable for generally decreased CD4 + naïve/Temra cells, and increased Treg and Tfh subsets. To look for T-cell signatures that may be shared across individual samples, we performed hierarchical clustering of all 107 (FL + rLN) samples based on relative abundances of T-cells across the 85 PG clusters. There were 3 main branches evident in the resulting dendrogram which were readily distinguished by their FL vs. rLN membership (Fig. 4C). One branch was composed mostly of rLN and included abundant naïve CD4 + and CD8 + cells (termed "naïve dominant"). Another contained mostly FL samples and was notable for increased Tregs with sub-branches rich in CD8 + effector memory (EM) and Th1 cells or Tfh cells (termed "CD8EM/Th-1-rich" and "Tfh-rich", respectively). The third branch included a mixture of FL and rLN samples appeared to be intermediate in cell composition between the other two branches (termed "mixed").

### Integration of B- and T-cell datasets
We next assessed co-occurrence of B with T cell populations using 107 samples for which both B-cell and T-cell CyTOF data were available

(73 FL and 34 rLN) and plotted the results in a force-directed map (Fig. 5A). Normal B-cells and various naïve and CM T-cell subsets formed a dense community that largely excluded tumoral B-cell clusters. The most populated B-cell PG cluster B01 (and major PG cluster in MC-A) significantly co-occurred with terminally differentiated (CD57 + ) subsets of Th1 and CD8EM T-cells (clusters T37 and T86, respectively; highlighted in Fig. 4A). The second most populated B-cell PG cluster B02 (and major PG cluster in MC-B) significantly co-occurred with a CD57-, cytokine-rich subset of Tfh T-cells (cluster T10; highlighted in Fig. 4A). This latter association between PG clusters B02 and T10 also extended more generally to the content of MC-B cells with total Tfh cells within each tumor (Fig. 5B). In contrast, the content of MC-A cells did not correlate with Tfh cells across samples. When tumors were classified into types A vs. B vs. NOS, Tfh cell content was significantly higher in each of the FL types as compared to rLN, while type B tumors contained significantly more Tfh cells than either of types A or NOS (Fig. 5C). These data confirm prior reports that Tfh cells are generally increased in FL[21,22], but additionally reveal an association with MC-B type FL cells in particular. Of note, immunohistochemical stains performed on type B tumors enriched for Tfh cells by CyTOF confirmed higher numbers of PD1 + T-cells within malignant follicles as

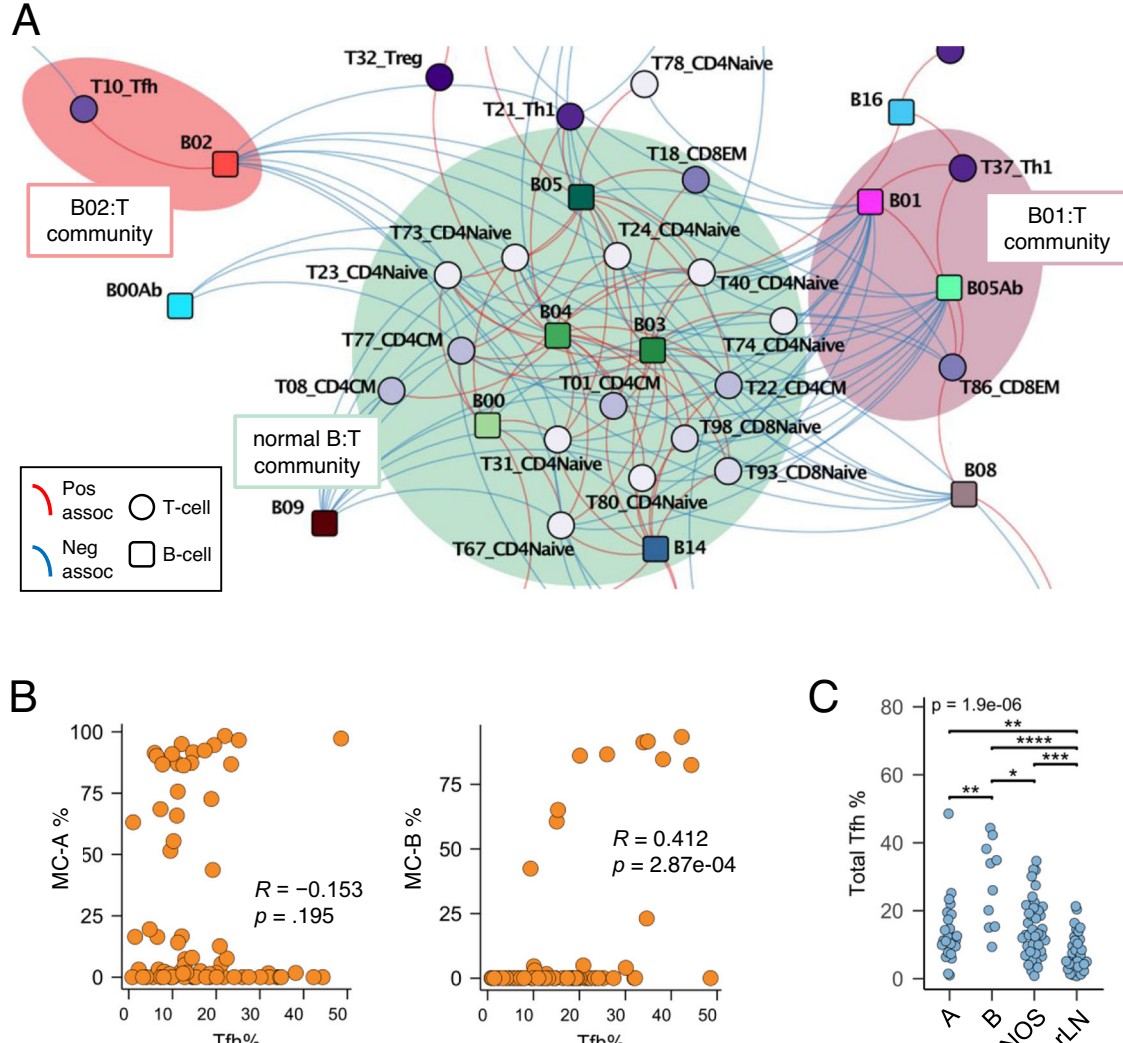

**Fig. 5 | B-cell types are associated with distinct subsets of T-cells. A** B/T-cell cluster co-occurrence network by force-directed graphing as in Fig. 3B. Probabilistic model-based co-occurrence analysis (2-sided test) was performed using B/T-cell cluster information from 73 FL and 34 rLN samples without additional adjustment for multiple testing (see Methods). Each node represents a B- or T-cell PG cluster. Each B-cell cluster is colored separately; T-cell clusters are colored according to landmark group. Communities local to normal B-cells and tumor B-cell clusters B01 and B02 are shown; see Supplementary Data 10 for the full network image. **B** Cell abundance correlation plots. Each dot represents an individual sample. B- and T-cell cluster % values are among total B- and T-cells in each sample, respectively. Spearman rank correlation rho (R) and p-values (2-sided) are indicated. **C** Tfh cell content by tumor type. Each dot represents an individual sample; Tfh % values are among total T-cells within each sample. Tumor type A, $n = 24$ samples; type B, $n = 10$ samples; type NOS, $n = 39$ samples; type rLN, $n = 34$ samples. *$p = 1.15e-02$; **$p = 6.20e-03$ (A vs. B) and $p = 5.18e-03$ (A vs. rLN); ***, $p = 1.75e-04$; ****, $p = 8.01e-07$ (Kruskal-Wallis test with post-hoc Dunn test). EM effector memory, CM central memory, Tfh T follicular helper, Th1 T helper type 1, Treg T regulatory, MC metacluster, NOS not otherwise specified, rLN reactive lymph node.

compared to type A tumors with low Tfh cell content by CyTOF (Supplementary Fig 15).

### FL subtypes exhibit distinct DNA mutation and rearrangement patterns

DNA mutational analyses from targeted panel sequencing (TarSeq) was available for 69 cases with similar proportions of A, B, and NOS tumors as in the full cohort (29/18.8/52.2% vs. 27.7/18.1/54%) (Fig. 6A and Supplementary Data 7). Mutations in *EZH2*, *MEF2B*, and *TNFRSF14*, were significantly increased in type A tumors as compared to all others (FDR *q*-values <0.1; Fig. 6B and Supplementary Data 7). No significant associations were noted for type B tumors among the genes on the TarSeq panel. These mutational associations will need to be verified in larger studies; however, they support the notion that phenotypic subsetting of MC-A type cells (which we use to define type A tumors) describes a distinct biological subset of FL. Of note, functional work in mouse models has shown that mutations in *EZH2*, *MEF2B*, and *TNFRSF14* confer growth advantage to GC B-cells[23-27], which would presumably correspond to the biology operative in MC-A type human FL cells.

Using RNA-Seq data available from 38 cases (Fig. S10), we also assessed somatic hypermutation (SHM) patterns, which revealed that tumor types A and B exhibited SHM of *IGHV* regions to similar extents (Supplementary Fig. 16 and Supplementary Data 8), supporting that both have previously transited through germinal center reactions. In combination with their respective phenotypes including surface IgM/IgG expression status (Supplementary Fig. 17), these results support that MC-A cells (which dominate within type A tumors) are best regarded as GC B-cells, while MC-B cells (which dominate within type B tumors) correspond more closely to pre-CSR but post-GC memory B-cells.

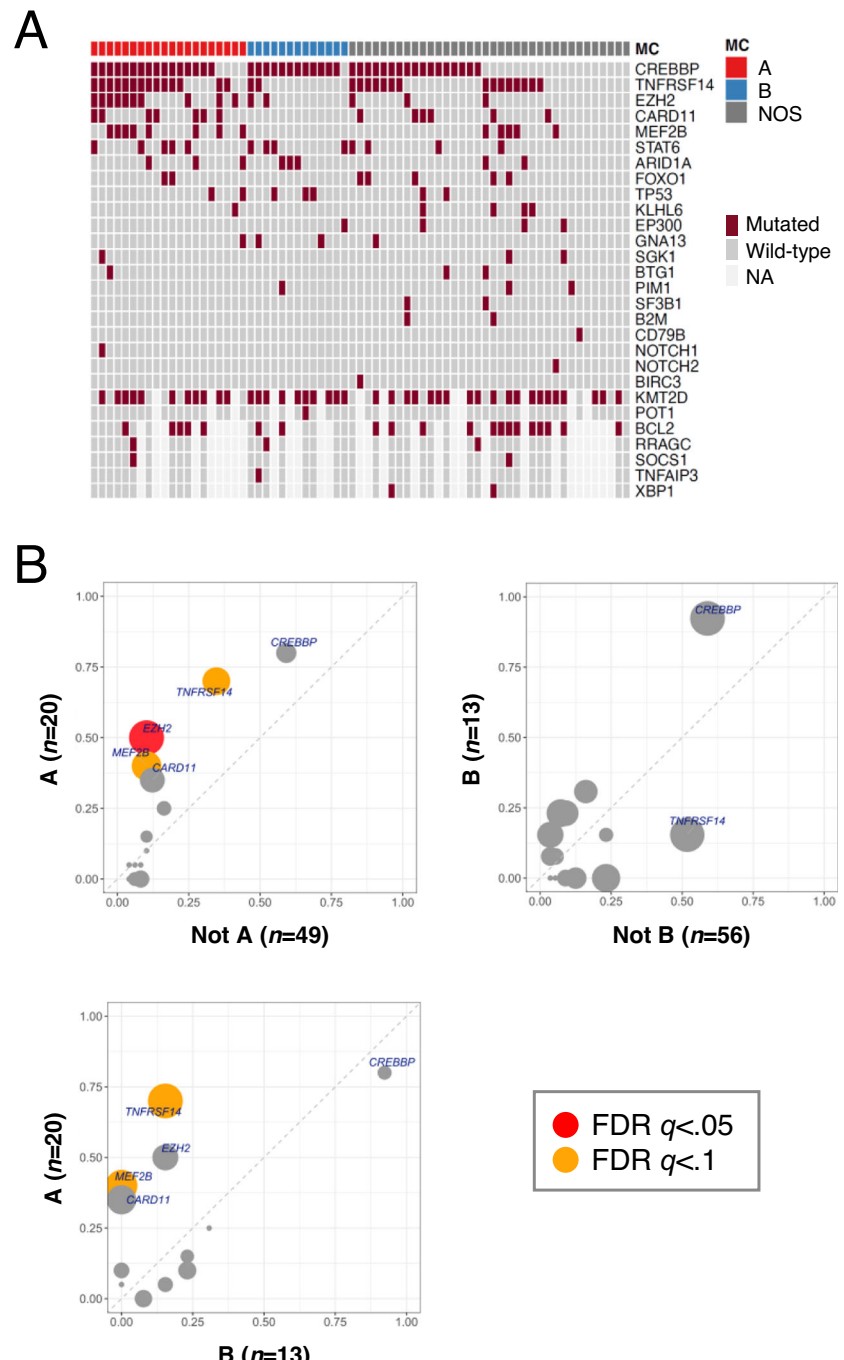

**Fig. 6 | FL subtypes show distinct DNA mutation patterns. A** DNA co-mutation plot from targeted sequencing of 69 FL samples. Tumor type is defined as >50% MC-A or MC-B cells among clonal B-cells, or neither. **B** Mutation frequency plots from targeted sequencing data (n = 69). Tumor types are as in **A**. Mutation bubble size is proportional to statistical significance (Fisher's exact test). MC metacluster, NOS not otherwise specified, FDR false discovery rate.

## FL subtypes and intratumoral entropy define patients with disparate clinical outcomes

We next examined whether there were any correlations between tumor phenotypes and clinical features at diagnosis. When parsed into the 2 major B-cell groups (types A and B) vs. all others (type NOS), performance status and stage were significantly different among the 3 B-cell groups with type B most enriched for poor PS and advanced stage (Supplementary Fig. 18a and Supplementary Table 2). When parsed into the 4 identified T-cell groups (naïve-dominant, CD8EM/Th1-rich, Tfh-rich, mixed), hemoglobin and stage were significantly different with the CD8EM/Th1-rich group most enriched for low hemoglobin and advanced stage (Fig.

S18b and Table S2). All other baseline characteristics were not significant.

We and others have previously examined DNA mutations to explore mechanisms underlying transformation in FL;[2,28] however, we considered here whether tumor phenotypes might be informative in this regard. Interestingly, types A vs. B vs. NOS showed significantly different risks of transformation (Fig. 7A) with type B showing the highest cumulative risk. Outcomes of individual MC types within the NOS group (i.e., C, D, E, F, etc.) were varied but difficult to assess due to limited numbers of these cases (Supplementary Fig. 19). We also looked at intratumoral phenotypic diversity among malignant B-cells (measured as entropy; Fig. 3C) and found it to be significantly

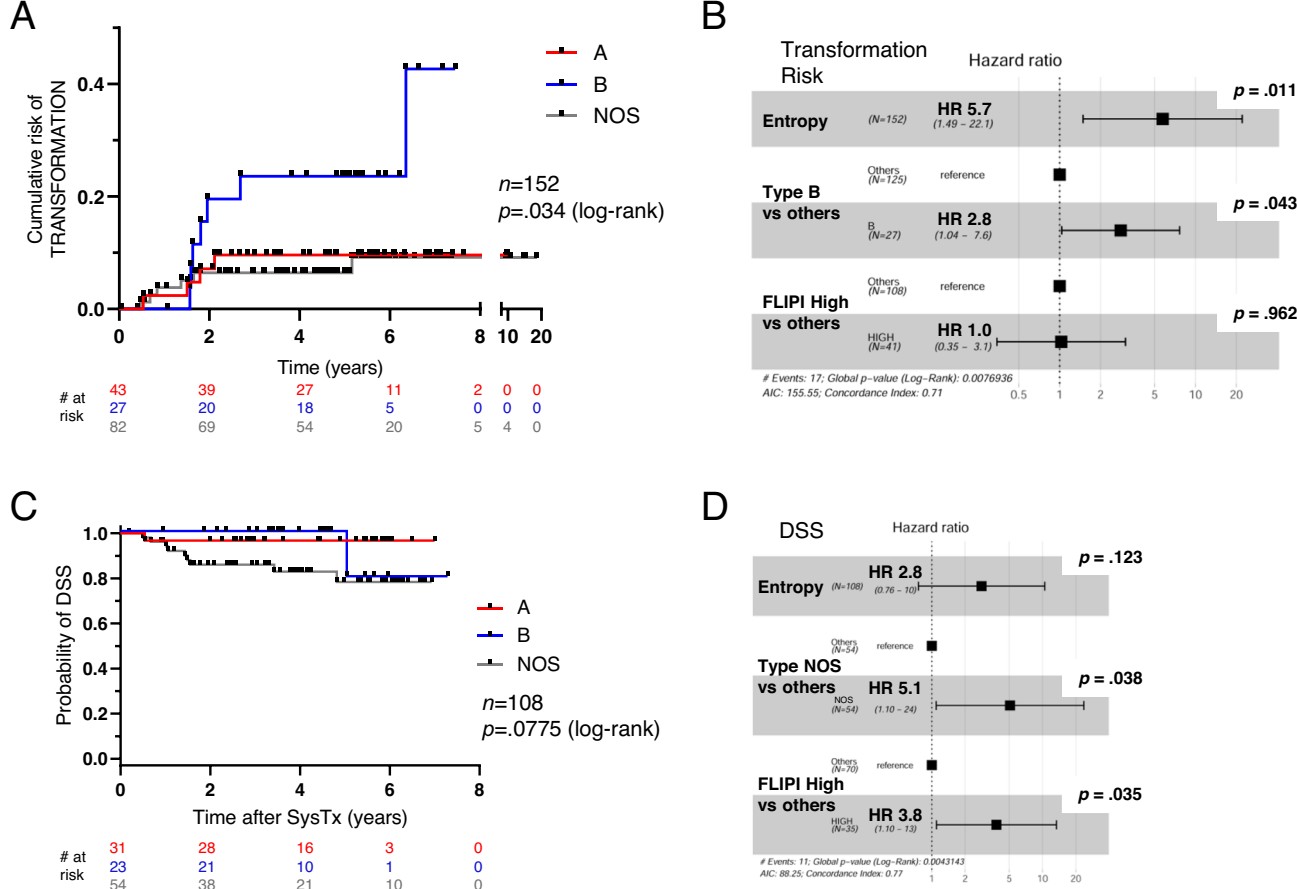

**Fig. 7 | FL subtypes show distinct clinical outcomes. A, C** Kaplan–Meier plots for **A** risk of transformation and **C** disease-specific survival (DSS). **B, D** Forest plots from multivariate analyses for **B** transformation risk and **D** DSS using Cox proportional hazards models. Hazard ratios (HR) are plotted with error bars indicating 95% CI. NOS not otherwise specified, DSS disease-specific survival, SysTx systemic therapy, FLIPI Follicular Lymphoma International Prognostic Index.

correlated with risk of transformation (Supplementary Fig. 20). In multivariate analyses with FLIPI group[29,30], both MC type B and entropy remained significant (Fig. 7B). As noted above, panel sequencing did not reveal any significant mutational associations with type B tumors, and this dataset was similarly non-informative for intratumoral entropy. Although more extensive sequencing may yet reveal mutational associations with MC-B and entropy, epigenetic features and/or tumor microenvironment may also contribute to phenotypic identity/ diversification in this setting[31].

To examine survival outcomes, a subcohort of 108 patients receiving a consistent primary systemic therapy regimen of bendamustine plus rituximab (BR) was identified with survival time calculated from start of systemic therapy. Of the 108 patients, 95 received BR up-front (median time to BR = 0.13 yrs), while 13 were observed prior to initiating BR therapy (watchful waiting; median time to BR = 6.1 years). Despite the increased risk of transformation associated with type B tumors, it was the NOS group that showed poorest outcomes for disease-specific survival (DSS) with MC type and FLIPI score as significant variables (Fig. 7C, D and Supplementary Fig. 21). It is worth emphasizing that the NOS group is a mixed bag of different MC types and likely subsumes multiple and diverse biologies which will require further study to delineate. Finally, there were no notable outcome associations among patients as grouped by the 4 T-cell signatures defined by compositional clustering (Fig. 4C).

Given that type B and high entropy tumors are more likely to transform, it remains unclear why these features are not associated with shorter survival. It should be noted however that the CyTOF cohort was relatively enriched for younger patients with larger tumors and who more often required primary systemic therapy as compared to the general FL patient population seen at our institution over the same time period (n = 992; Supplementary Table 1). Accordingly, further work will be needed to determine if these clinical associations bear out in independent and more representative patient cohorts.

## Discussion

A widely held view of FL pathogenesis is that tumors arise from B-cells following iterative cycles of GC-re-entry with SHM and CSR providing the mutational drive for clonal establishment/progression[32]. Viewed in this context, GCB and MB tumor types as described here could reflect developmental arrest at different points in the re-entry cycle, i.e., within and outside of the GC proper, respectively. The developmental stage of MB type cells appears to correspond most closely to pre-CSR memory B-cells but could potentially also encompass so-called "FL-like cells" (FLLC) which paradoxically harbor DNA rearrangements involving IGH switch regions yet maintain surface IgM expression[33–36]. Identification of this alternate, memory-like cell type in FL and its associated increased risk of transformation suggests that important aspects of the lymphomagenic process may occur outside of the GC proper. Interestingly, recent work has shown that CSR may indeed occur prior to GC entry[37,38], in which case MB type cells could potentially be subject to CSR-induced recombination at greater levels than GCB type cells. The observation that Tfh cells correlate with MB rather than GCB type FL cells and are found in greatest abundance in type B tumors (Fig. 5) would also fit with an increasingly recognized role for Tfh cells outside of the GC proper[39].

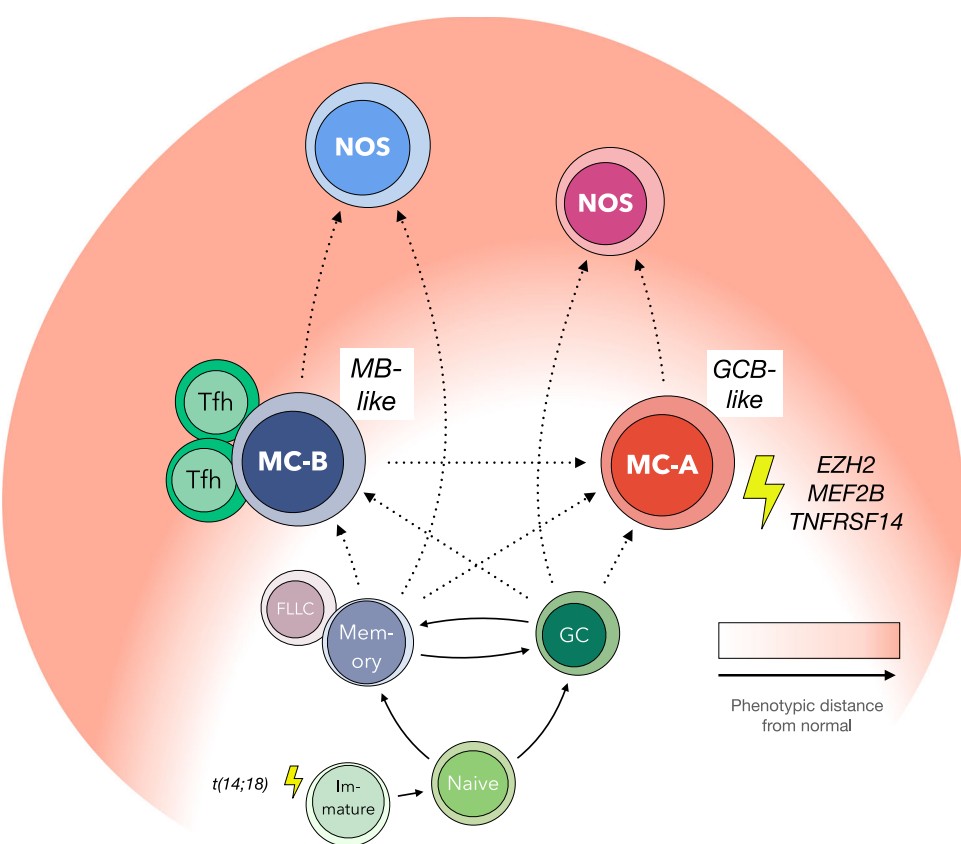

**Fig. 8 | Hypothetical model of FL genesis.** MB memory B-cell, GCB germinal center B-cell, MC metacluster, NOS not otherwise specified, Tfh follicular helper T-cell, FLLC follicular lymphoma-like cell.

In comparison to prior studies of patient FL samples performed at single cell resolution, our type MC-B may correspond to the naïve/memory type observed by Wogsland et al.[40]. When viewed in the context of functional plasticity as proposed by Milpied et al.[41], one interpretation of our data could be that the GC-like MC-A and memory-like MC-B types represent interconverting or dynamic functional states as opposed to distinct, static phenotypes. The CyTOF data presented indeed captures only a snapshot in time of each patient's disease; however, co-occurrence of MC-A and MC-B cells in the same sample is statistically underrepresented, thus arguing against the notion that FL tumor cells actively interconvert between these two types. Moreover, type MC-A cells are typically IgG+ and deletion of the intervening Cμ DNA segment would presumably prevent reversion back to an IgM+ state as is typically seen in type MC-B cells. Taken together, these features suggest that MC-A cells likely do not give rise to MC-B cells; however, the data do not exclude the possibility that MC-B cells could give rise to MC-A cells but diminish rapidly from the tumor thereafter. We would thus conclude that functional plasticity of the sort described by Milpied et al. would most likely represent a separate phenomenon from the GCB vs. MB tumor cell types described here.

The unexpected contrasts between the two most common FL cell types MC-A and MC-B (phenotypes, infrequent co-occurrence, associated T-cells) combined with distinct mutational associations and transformation risks suggests they may reflect distinct biologies. These two cell types could potentially arise independently from one another (i.e., from alternate GCB- or MB-like stages of differentiation), arise from a common precursor (yet deviate from one another under the influence of subsequently acquired gene mutations, epigenetic modifications, or microenvironmental interactions), or develop in sequential order (i.e., MC-B giving rise to MC-A). Perhaps the most

likely situation is that all possible paths are explored by tumor cells as they develop and progress within constantly evolving genetic/epigenetic, cell developmental stage, and environmental contexts. Finally, since about 20% of tumors show no detectable MC-A or MC-B cells (Fig. 3D), tumors arising independently of MC-A or MC-B phenotypes cannot be excluded.

FL tumors dominated by cells of other, more phenotypically divergent MC types (e.g., MC-C/D/E/F/etc.) often contained small proportions of MC-A or MC-B cells (Fig. S22). Also, type MC-C cells tended to co-occur more frequently with MC-A cells while types MC-D/E/F/Mem/Nav co-occurred more frequently with MC-B (Fig. 3A, B). One interpretation of these observations could be that FL tumors trace ontogenic paths involving MC-A and/or MC-B types initially, and then progress onwards to more divergent (NOS) phenotypes (Fig. 8). Type NOS cases could thus potentially be regarded as further along in the natural history of disease, thereby suggesting a possible explanation as to their apparently shorter DSS (Fig. 7C, D). Marked phenotypic heterogeneity across MC types within the NOS group (Fig. S9) suggests they represent a diverse collection of biologies, however, and not a distinct entity per se with poor survival. Larger numbers of NOS cases will need to be studied to determine if there is any biological commonality among those with shorter survival.

An alternate possibility could be that tumors progress towards, rather than away from MC-A and MC-B phenotypes over time. One scenario could be, for example, that evolving tumor cells are funneled into normative phenotypes as they adapt to a limited range of supportive lymph node microenvironments. These notions are admittedly speculative, however, and would need to be evaluated by dedicated approaches to determine if there exists a hierarchy of mutations within different phenotypic subsets of a given tumor, or if other aspects such

as the epigenome or tumor microenvironment may be responsible for creating the observed patterns of cell phenotypes.

It is worth noting that while 80% of transformed FL are DLBCL of GCB type[31], our data would support that type B (memory B-like) FL have higher risk of transformation than type A (GCB-like). In considering this ostensible disconnect, we would point out that type B comprises under 20% of FL at diagnosis, and that while pre-switch IgM + type B tumors could presumably transform to either ABC-DLBCL (which are most often IgM/D + ) or GCB-DLBCL (which are most often IgG/A + )[42], post-switch IgG+ type A tumors would presumably more likely give rise to an IgG+ GCB-DLBCL than an IgM+ ABC-DLBCL (at least directly). Further, *TNFRSF14* and *EZH2* mutations are seen much more frequently in GCB- than ABC-DLBCL[43,44], and would imply that type A FL, which are also enriched for these mutations, would be predisposed to adopt GCB/EZB/C3 character upon transformation. In contrast, pre-switch IgM+ type B FL with no clear mutational associations would presumably have a wider range of transformation paths available. Finally, our prior study examining clonal dynamics in progressed vs. transformed FL has shown that the overt genetic signature of FL changes dramatically after transformation[2], and thus it may be misleading to draw rigid connections between pre- and post-transformation tumor profiles.

While additional studies will be needed to validate clinical outcome associations with tumor MC type and entropy features, these may not necessarily require that CyTOF be performed to determine tumor cell phenotypes. In fact, ranking of individual markers by their contribution to discriminating MC types A vs. B vs. NOS revealed that 98% of information content is captured with just 26 of 39 markers from the full CyTOF panel (Fig. S23 and Supplementary Data 9) and re-analysis of the CyTOF data using just these top 26 markers largely retained the ability to identify high-risk FL cases in terms of type B and entropy features (Fig. S24). Of note, segregation of tumors by IgM/IgG status alone did not reveal significant differences in clinical outcome (Fig. S25). Validation of these findings in additional cohorts is needed prior to consideration for development as a clinical assay; however, a reduced 26-marker set is at least within reach of conventional flow cytometry and thus could reasonably be deployed in some clinical settings.

## Methods

### Patient samples
Excess single-cell suspension material from excisional LN biopsy specimens remaining after clinical diagnostic assessment were prospectively banked with DMSO cryoprotectant. Informed consent or consent waiver was obtained for all samples utilized for research according to protocols approved by the University of British Columbia/BC Cancer Agency Research Ethics Board.

### Antibody staining
Cryopreserved cell suspensions were thawed at 37 °C, washed in complete media (RPMI-1640 + 10% FCS), and stained with B- and T-cell antibody panels (Supplementary Data 9). Antibodies not already conjugated to metal tags by the vendor were conjugated in-house using Maxpar X8 Multimetal Labeling Kits (Fluidigm cat# 201300).

For B-cell panel staining, cells were first incubated with 25uM cisplatin in serum-free media to label dead cells. Cells were next stained with any antibodies against antigens sensitive to the barcoding procedure, then each sample was barcoded using palladium-based mass tags (Cell-ID 20-Plex Pd Barcoding Kit; Fludigm cat#201060). After sample barcoding, cells from up to 12 different samples were pooled into a single tube and stained with the remaining panel antibodies in bulk.

For T-cell panel staining, thawed cells were transferred immediately onto 96-well plates and rested overnight at 37 C in a humidified, 5% CO2 incubator. The following day, cells were stimulated for 4 h with PMA (200 ng/ml)/Ionomycin (1.5ug/ml) in the presence of Brefeldin A (2 μg/ml) and monensin (1 μg/ml) (GolgiStop cat#554724 and GolgiPlug cat#555029; BD Biosciences), then stained sequentially with cisplatin, barcoding-sensitive antibodies, and Pd barcodes. Cells from up to 12 different samples were then pooled into a single tube, treated with Cytofix/Cytoperm Fixation/Permeabilization Kit (BD Biosciences cat#554714), and stained with the remaining antibodies.

Each of the 12-plex barcoded sample batches included an aliquot of cells from a master pool of 10–20 rLN samples to serve as an internal staining control and enable batch-to-batch normalization. After antibody staining was completed, cells were fixed with methanol and stained with Cell-ID Intercalator-Ir dye (Fluidigm cat#201192) and prepared for CyTOF acquisition according to the manufacturer's protocols. Cells were acquired on a CyTOF2 instrument (Fluidigm) equipped with Super Sampler (Victorian Airship). Typically, ~50,000 live B- or T-cell events per sample were obtained.

### CyTOF data pre-processing
FCS files from each acquisition batch were concatenated using CyTOF Software (v6.0.626, Fludigm) and normalized with spiked-in EQ Four Element Calibration beads (Fludigm cat#201078) using Normalizer (v0.3)[45] from MATLAB (v8.6). Bead-normalized FCS files were then imported to MATLAB-based Single-cell Debarcoder (https://github.com/zunderlab/single-cell-debarcoder)[46] which generates sample-assigned FCS files with two debarcoding parameters (separation and mahalanobis distances). Sample-tailored 2-D gates were created based on these two parameters in FlowJo (v10; BD Biosciences) to achieve optimal cell yield while maintaining high specificity for each sample. For B-cell panel analysis, viable non-T cells were gated in FlowJo based on negative staining for cisplatin and CD3, then exported as FCS files. DNA gating was not applied in these analyses since we observed DNA-gain in some FL samples and further that single-cell debarcoding procedures efficiently removed cross-sample cell doublets. For T-cell panel analysis, viable T-cell singlets were gated and exported as FCS files. All downstream analyses were subsequently performed in R (v3.3-v3.5) or Python (v3.6).

### Batch effect normalization and compensation
Cell-derived normalization using the pooled rLN control was applied to correct for batch-to-batch variation[47]. Briefly, channel-based normalization factors were calculated by dividing the median intensity of each channel from each batch with weakest median intensity of corresponding channel among all batches (normalized to the weakest signals across all batches). Normalization of each sample was performed by dividing the expression matrix by a vector containing batch-specific normalization factors for each channel. We also assessed spillover/crosstalk from channels 142Nd, 155Gd, 160Gd, 162Dy, 163Dy, 172Yb, and 174Yb using spillover controls. Spillover in the actual samples was corrected using the CATALYST (v1.10.3) package with non-negative least-squares (NNLS) method to avoid introducing negative values[48]. Normalized and compensated files were transformed using inverse hyperbolic sine (arcsinh) function in FlowCore (v1.52.1) package ($a = 0.2$, $b = 0$).

### Dimensional reduction
We chose UMAP dimensional reduction tool[6] from umap-learn (v0.3-v0.4) package to visualize single-cell data with the following settings: minimal distance (md) =0.4 and nearest neighbor number (nn) = 30. t-SNE was also used in some analyses, mostly for QC assessment. We used the Barnes-Hut Stochastic Neighbor Embedding (bh-SNE implementation)[49] available in Rtsne (v0.15) R package. Markers used for dimensional reduction and clustering analyses are indicated in Supplementary Data 9. We replaced the two Ig light chain markers, kappa and lambda, with a single anonymized Ig light chain marker, "KL", using the higher value from either kappa or lambda.

## Unsupervised clustering of B-cell data

To accommodate computational limitations, typically ~5000 non-T cell (CD3-negative) events were randomly subsetted from each patient sample, then concatenated into a single data matrix containing ~900,000 total cell events. We then performed clustering using the graph-based algorithm PhenoGraph (v1.5.2; Python package) on the same data matrix as for dimensional reduction with nn = 100, which was used previously on a dataset of similar size[50]. To minimize stochasticity of the clustering algorithm, we first performed 25 iterations of PhenoGraph clustering. We then then applied the Adjusted Rand Index (ARI)[51] and Normalized Mutual Information (NMI)[52] measures to generate pair-wise similarity scores and plotted the resulting 25 sets of 24 values each. We defined the consensus cluster assignment as the set of results with highest average ARI and NMI scores (ARI: 0.96, NMI: 0.95). ARI and NMI scores were calculated using functions from aricode (v1.0.0) R packages. To reduce noise from rare cell events, we retained only those cells that were assigned to Phenograph clusters containing at least 1% of total non-T cells in each respective sample. Following Phenograph clustering, all output clusters containing verifiable CD19+, CD20+, and/or CD22 + B-cells were meta-clustered by hierarchical clustering using the hclust function in R. We then selected 19 meta-clusters as optimal based on the gap statistic[8].

## Inter-sample entropy calculation

We used Shannon entropy to help identify phenotypically similar cells across samples. For a dataset with $n = 191$ samples, a K-NN graph ($K = n$ −1) was constructed to find the $K$ nearest neighbors, followed by calculating the proportion ($p_m$) of each sample among the $K$ nearest neighbors. Then we calculated the inter-sample entropy score for each cell as $-\sum_{m=1}^{190} p_m \cdot \log(p_m)$

## Intra-tumoral entropy calculation

For each sample we first calculated $p_i$, as proportion of every abnormal cell type or PG (i = 1,...,n) among its total abnormal cells, entropy scores were then calculated as $-\sum_{i=1}^{n} p_i \cdot \log(p_i)$

## Unsupervised clustering of T-cell data

To accommodate computational limitations, we clustered 856,000 T-cells from 107 samples (8,000 cells randomly subsetted per sample) using multi-level PhenoGraph (nearest neighbor number, k =100) as implemented in iGraph (v1.2.6). Markers included in the clustering analysis are indicated in Supplementary Data 9. The top layer of clustering results which yielded 104 clusters was extracted. After pruning rare clusters, the 85 remaining clusters were used for downstream analyses.

## Scaffold map analysis

To visualize large numbers of T-cell subsets identified by unsupervised clustering and to interrogate differences in T-cell content between FL and rLN samples, we employed the Scaffold Map Analysis approach[53,54]. Scaffold mapping organized unsupervised clusters (see above) together with 11 manually curated conventional landmark T-cell populations (Tem = CD197- CD45RA- CD45RO +, Tcm = CD197 + CD45RA- CD45RO +, naïve = CD197 + CD45RA + CD45RO-, Temra = CD197- CD45RA + CD45RO-, Treg = CD4 + CD127- CD25 +, Tfh = PD1 + CXCR5 + CD25-, Th1 = CD197- IFNγ + ; Fig. S14) to provide visual cues when exploring the landscape of T-cell populations. Each landmark node was allowed to keep up to 20 edges, and each sample node up to 10 edges. Force-directed maps were generated in Gephi (v0.9.2) using the ForceAtlas algorithm. Since TIM3 and LAG3 antibody staining in the first 5 acquisition batches was not optimal, these markers were excluded from initial construction of the Scaffold map; however, in the final steps of map construction, each node was assigned with median values for TIM3 and LAG3 which were calculated based on data from

corresponding nodes acquired in the latter 7 batches. Significance Analysis of Microarrays (SAM v3.0)[53] was performed to identify significant changes in the abundance of cells within each node between FL and rLN samples. The SAM method employs a permutation-based approach to control for Type I errors and accordingly reports FDR values. Each sample node was also classified into one of the landmark groups based on highest pairwise similarity.

## Co-occurrence analyses

To explore the co-occurrence of B-B, T-T, and B-T populations within individual samples, we employed a probabilistic co-occurrence model originally designed for ecological studies into the role of species coexistence in community structure (cooccur v1.3)[20]. We assigned values of 1 to indicate presence of those PG-defined populations with abundances greater than 1% of total viable cells within each sample, and values of 0 to indicate absence for those with less than 1% abundance. We then defined the probability of presence for each population simply as the number of patient samples in which the population was present over the total number of samples. Probability distributions for co-presence of all possible pairs of populations were calculated and observed co-occurrence frequencies were then compared against these distributions to determine which, if any of the co-occurrences were statistically significantly increased or decreased (alpha = 0.05). Additional adjustment for multiple testing was not performed as the numbers of possible cluster pairs and total samples did not together exceed those modeled using simulated random data to assess for Type I errors in the originally reported method[55]. The ForceAtlas algorithm in Gephi (v0.9.2) was then used to graph significant positive and negative associations between nodes. Since ForceAtlas does not accept negative values, association scores were transformed using the exponential function, $e$.

To assess mutual exclusivity of MC-A and MC-B type cells within individual samples, we examined all available data events up to 50,000 non-T cells per sample. We applied the 1% threshold for presence/absence as above but calculated as a fraction of total malignant B-cells within the sample. Similar results were obtained when the cutoff was reduced down to 0.5, 0.4, and 0.3%, or alternatively, down to an absolute number of 50 cells.

## Immunohistochemistry

Formalin-fixed, paraffin-embedded (FFPE) serial whole tissue sections (4 um thickness) from diagnostic FL biopsies were stained with antibodies against CD3 (polyclonal, Dako cat#GA50361-2), CD57 (clone TB01, Dako cat#GA64761-2), or PD1 (clone NAT105, Cell Marque cat#315 M) after antigen retrieval at 97C with high pH for 20 min on a Dako Omnis automated slide-staining system. Whole slide images were acquired using a MoticEasyScan Pro digital slide scanner (x40 magnification, standard mode) and viewed using Aperio ImageScope (v12.4.3.5008) software.

## RNA-Seq

Single-cell RNA-sequencing (scRNA-Seq) was performed on the 10x Genomics platform with Chromium Single Cell 3′ Chip Kit v2 (10x Genomics cat#1000009). Libraries were constructed using the Single Cell 3′ Library and Gel Bead Kit v2 (10x Genomics cat#120237) and Chromium i7 Multiplex Kit 10x Genomics cat#120262). Two single-cell libraries were pooled and sequenced per HiSeq 2500 125-base PET lane.

Conventional (bulk) RNA-sequencing was performed on unfractionated cell suspension or snap frozen whole tissue material. Total RNA was isolated with TRIzol reagent followed by purification over PureLink RNA Mini Kit columns (Invitrogen cat#12183018 A). RNA-seq was performed using a polyA-enriched strand-specific library construction protocol[56] and paired-end 75 bp sequencing on an Illumina HiSeq 2500 instrument.

## Single-cell RNA-Seq data analysis

CellRanger software (v2.1.0) was used to demultiplex the raw data, generate quality metrics, and generate per-gene count data for each cell. Data was then imported into the R statistical environment (v3.5.2) as SingleCellExperiment objects. Scater (v1.8.0) R package was used for data pre-processing and quality filtering. A total of 31,026 cells across the 6 FL + 4 rLN samples, or ~3,000 cells per sample were recovered after data pre-processing and used for analysis. Count data was log-normalized and Scran (v1.9.11) R package was used for batch correction (fast mutual nearest neighbors, MNN). The matrix containing corrected low-dimensional coordinates for each cell was used for downstream analyses.

For visualizing these data in UMAP, we used the pl.umap function from Scanpy (v1.6.0). We used network-based Phenograph clustering including pp.neighbors to find nearest neighbors, followed by tl.louvain to define the clusters. Clusters were annotated with normal B-cell subset designations where appropriate based on results from a recent scRNA-seq study of normal human B-cells[37]. Hierarchical clustering of samples and normal B-cell subsets was performed using the scanpy.tl.dendrogram function with default settings. To identify RNA species most correlated with CyTOF-defined type A vs. type B tumor samples, we used the rank_genes_groups function with Wilcoxon rank-sum method (two sided) followed by Benjamini-Hochberg correction for multiple testing ($p < 0.05$) and then filtered for genes with log2 fold-change >1.

## Bulk RNA-Seq data analysis

Raw reads were aligned to the reference human genome assembly GRCh37 (hg19) using STAR (v2.5.2.a). To improve spliced alignment, STAR was provided with exon junction coordinates from the reference annotations (Gencode v19). We applied a modified version of a bioinformatics workflow for normalization of raw read counts and differential gene expression analysis[57]. Gene-level read counts were quantified using HTSEQ-count (v0.11.0; intersection-strict, reverse mode)[58]. Genes showing low read counts (i.e., genes not showing counts per million (cpm) > 1.0 in at least 10% of samples) were removed from further analysis. Raw counts from expressed genes were then TMM-normalized and scaled to counts per million (CPM) using the edgeR (v3.22.2) package[59].

Differential gene expression analysis was performed in R (v4.1.1) using the DESeq2 package (v1.34). Batch correction was performed to account for sample source (cell suspension vs. snap frozen whole tissue) using the removeBatchEffect function in limma (v3.50.3). Differentially expressed genes were filtered for absolute log2 fold-change > 1 and Benjamini-Hochberg-corrected $p$-value <0.05 (2-sided Wald test). Heatmaps were generated using the pheatmap package (v1.0.12).

## Targeted sequencing

We used the TruSeq Custom Amplicon assay (TSCA; mean coverage: 767; range: 128–2,039; SD: 180) to identify variants within the protein coding regions of 59 genes commonly mutated in human B-cell lymphomas (Supplementary Data 7)[60–62]. TSCA variants were validated with the Fluidigm Access Array system which achieved a 97% validation rate. Discrepancies between TSCA and Fluidigm results were further validated by Sanger sequencing.

## Targeted sequencing data analysis

Reads were mapped with BWA (v0.7.5a). SNVs and indels were predicted with Mutascope (v1.02). SAMtools (v0.1.19) was used to create pileup files and dbSNP (v137) for SNP annotation. All variants with an allele frequency of ≥5% at loci covered at least 50-fold were retained.

## IGHV mutation status assessment

To identify the immunoglobulin heavy-chain variable (*IGHV*) mutation status from bulk RNA-seq data, we developed an in-house pipeline[63] motivated by the work of Balchly et al[64]. Briefly, we performed de novo assembly of the paired-end RNA-seq reads using Trinity (v2.1.1)[65] to construct *IGHV* transcripts. We selected the most abundant and productive *IGHV* transcript to define somatic hypermutation status. The selected transcript sequence was queried against NCBI IgBLAST (v1.14.0) to identify *IGH-V*, *D*, and *J* genes. IgBLAST was used with default parameters to detect percent identity between the query and the highest similarity germline *IGHV* gene.

## Statistical analyses

All statistical tests were performed as two-sided tests where applicable using R packages (v3.3-v3.5 & v4.1.1) or Prism v8 (GraphPad).

## Clinical outcome analyses

We defined transformation based on biopsy-confirmed histology consistent with diffuse large B-cell lymphoma or clinical criteria as outlined previously[1] where at least one of the following were present: sudden rise in LDH to greater than or equal to twice the upper limit of normal, rapid discordant localized nodal growth detected clinically or by imaging studies, new involvement of unusual extranodal sites, new B symptoms, or development of new hypercalcemia. Non-parametric Kaplan-Meier (KM) survival, semi-parametric Cox regression, and weighted Cox regression analyses where necessary were performed. While there were no significant non-linear patterns in the survival models, the constant hazard ratio (HR) assumption did not hold for all models. We applied the regular Cox model for those with constant HR; otherwise, a weighted Cox model with estimated average HR was used.

## Reporting summary

Further information on research design is available in the Nature Research Reporting Summary linked to this article.

## Data availability

Source CyTOF datafiles are available on FlowRepository under accession #FR-FCM-Z3EL. These data are associated with Figs. 1, 3, 4, and 5. scRNA-seq BAM files (generated with CellRanger v2.1.0) for the 4 rLN samples have previously been deposited in the European Genome-phenome Archive (EGA) under accession #EGAS00001004085[66]. scRNA-seq BAM files for the 6 FL samples have been deposited into EGA under accession #EGAS00001005257. Access to these data is restricted to academic use only due to patient privacy concerns relating to potentially identifiable sequence-level information. Access can be requested from the Data Access Committee via the EGA portal with data made available within approximately 2 months. These data are associated with Fig. 2. Bulk RNA-seq FASTQ data files have been deposited in the EGA under accession #EGAS00001006646. This data is part of an ongoing study, and is also available under restricted access. Access can be requested as above. Genome alignments were performed against the reference human genome assembly GRCh37/hg19 [https://www.ncbi.nlm.nih.gov/data-hub/genome/GCF_000001405.13/]. Exon junction coordinates were referenced from GENCODE release 19 [https://www.gencodegenes.org/human/release_19.html]. Single nucleotide polymorphisms were identified using dbSNP build 137 [https://www.ncbi.nlm.nih.gov/projects/SNP/snp_summary.cgi?view+summary=view+summars&build_id=137].

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

## Acknowledgements

This work was supported by operating grants from the Cancer Research Society (Montreal; to A.P.W.) and Canadian Institutes for Health Research (CIHR; to A.P.W.), a Program Project Grant from the Terry Fox Research Institute (TFRI; to A.P.W., S.P.S., C.St, and D.W.S.), Large Scale Applied Research Project funding from Genome Canada, Genome BC, and CIHR (to C.St and D.W.S.), and infrastructure support from the BC Cancer Foundation.

## Author contributions

X.W., M.N., D.G., M.K., G.S., E.A.C., G.C.S., and J.K. generated data. X.W., M.N., G.D., C.Sa, L.H., R.W., T.A., R.I., C.M., S.H., K.T., R.D.M., and A.J.R. analyzed data and interpreted results. A.J. performed survival analyses. C.Sa, T.A., C.F., L.H.S., and K.J.S. provided clinical information and insight. R.R.B., A.K., and M.H. provided advice. J.W.C. and A.P.W. reviewed histology. S.P.S., C.St, D.W.S., and A.P.W. conceived the study and provided project supervision. X.W. and A.P.W. wrote the manuscript.

## Competing interests

The authors declare no competing interests.
