## [Peer Review File · Nature Communications]

This manuscript has been previously reviewed at another journal that is not operating a transparent peer review scheme. This document only contains reviewer comments and rebuttal letters for versions considered at *Nature Communications*.

REVIEWER COMMENTS

Reviewer #4 (Remarks to the Author): Clinical lymphoma expertise

These authors report descriptive studies in which CyTOF defined B-cell subsets within viably preserved FL tumors. Two groups of tumors emerged enriched for particular clusters with similarities to germinal center or memory-B cells, while other cases were enriched in other clusters or were more heterogeneous. The memory cluster had higher risk for transformation to tFL and differential mutation patterns from the GCB cluster. The report provides extensive data that is informative to the FL basic research field, though for now challenging to translate clinically given the techniques employed. There are some confusing aspects of the data and conclusions the authors should address prior to publication.

-Transformation of FL is almost always to GCB-DLBCL (Cluster 3/EZB by Chapuy/LymphGen). This project concludes, however, highest risk for transformation is in a memory-B subset specifically lacking defining Cluster3/EZB features. The authors need to reconcile this apparent contradiction. (From prior review, the reviewer questioned claims regarding whether different cells of origin are truly established by these data, which can be seen as another way of stating this point.)

-Use of anthracycline-based chemo for tFL does not to my knowledge eliminate the poor prognostic impact of transformation compared to non-transformed cases. To what extent might selection bias have impacted survival outcomes in this set of patients? (i.e. are cases with available viable samples different from the general FL population?) This comparison could be performed I believe with data available to these authors.

-Is there any evidence for an enrichment in Tfh cells in tFL cases?

-I was asked also to consider prior review before transfer of the ms to Nat Comms. Some points overlap with above:

-I agree with that reviewer that conclusions regarding implications of this study should be altered. Functional studies and validation in additional cohorts for now are the only future directions with no current avenue for clinical translation therapeutically or for establishment of clinical biomarker assays.

-Detailed clinical comparison of these cases to overall FL (as above) can help address some questions of robustness.

Minor points:

-How do the authors justify 50% of cells in A or B clusters to assign that case to that subtype? Was this chosen arbitrarily or based on statistical calculations that compared heterogeneity of FL populations across tumors?

Reviewer #5 (Remarks to the Author): expertise in single cell analysis in lymphoma

Wang et. al. present single-cell phenotyping of a large Follicular Lymphoma cohort. They identify a subset of FL tumors with germinal center B cell phenotype associated with an increased risk of transformation. Given that a subset of FL patients progresses to aggressive disease over time, this is a highly relevant study for the field. The cohort size and phenotyping based on multiple assays represents a unique dataset of great value. My concerns regarding the single-cell analysis are as follows:

Major:

1. This is a very difficult story to follow given its current structure. One of the factors that causes the most confusion is switching between phenograph clusters (PGs) and meta clusters (MCs) when describing the B cell results.
2. Lines 137-143: 54% of samples are non-A, non-B based on the thresholds used for classification. We are provided little information on what their phenotypes are. Figure S10 is dense and difficult to interpret. Markers in table S4 contain CD44, CD24, Igs that are also considered to be characteristic of A,B subtypes. In Fig. 7C, nonA-nonB tumors had the worst survival outcome. It

seems like a highly prevalent and important category of tumors that should not be ignored.

3. It seems like a missed opportunity that only a superficial analysis from scRNA-seq is presented. The granularity of this assay has the potential to clearly delineate (i) unbiased biology of what distinguishes type A and B tumors, (ii) extent of intrasample heterogeneity including non-A, non-B features. Which samples were used for scRNA-seq analysis? From Fig. S14, it seems like they were independent from the CyTOF samples? In which case: how were these samples classified into A/B? If they overlapped, what % of their cells were found to be A/B/nonA-B based on CyTOF analysis? A wealth of information could be gained from an unbiased analysis of these results rather than using them to validate markers from CyTOF.

4. Fig. 6A mutations seem to overlap in A,B, non A-B. However, Fig. 6B is presented as A vs rest, B vs rest. This makes the results seem more mutually exclusive than they are.

Minor:

1. Conclusions are sometimes over-interpreted. Example Line 174: ..."suggesting that phenotypic variation within individual tumors tends to occur in a progressive, incremental manner."

2. Perhaps Fig. S14 should appear much earlier in the paper to give us an idea of what assays were performed and on which samples.

Reviewer #4 (Remarks to the Author): Clinical lymphoma expertise

These authors report descriptive studies in which CyTOF defined B-cell subsets within viably preserved FL tumors. Two groups of tumors emerged enriched for particular clusters with similarities to germinal center or memory-B cells, while other cases were enriched in other clusters or were more heterogeneous. The memory cluster had higher risk for transformation to tFL and differential mutation patterns from the GCB cluster. The report provides extensive data that is informative to the FL basic research field, though for now challenging to translate clinically given the techniques employed. There are some confusing aspects of the data and conclusions the authors should address prior to publication.

-Transformation of FL is almost always to GCB-DLBCL (Cluster 3/EZB by Chapuy/LymphGen). This project concludes, however, highest risk for transformation is in a memory-B subset specifically lacking defining Cluster3/EZB features. The authors need to reconcile this apparent contradiction. (From prior review, the reviewer questioned claims regarding whether different cells of origin are truly established by these data, which can be seen as another way of stating this point.)

>> Thank you for raising this interesting point. Our own published work on FL transformation has documented that while indeed 80% of FL transform to GCB-DLBCL, 16% do transform to ABC-DLBCL (n=110; doi.org/10.1182/blood-2015-06-649905). When considering that memory B-like FL (type B) cases constitute only ~18% of FL, it seems reasonable to posit that these cases may preferentially transform to ABC rather than GCB-DLBCL. Of note, Ruminy et al showed that ABC/GCB types correlated significantly with BCR isotype where 24/26 ABC cases expressed IgM and 15/17 GCB cases expressed IgG or IgA (doi.org/10.1038/leu.2010.302). Since class switch recombination (CSR) is irreversible, it would follow that IgM+ ABC-DLBCL are much more likely to have arisen from an IgM+ FL (as are type B cases) than from an IgG+ FL (as are type A cases). It is of course also possible that an IgM+, type B FL could undergo transformation, and along the way class switch to IgG and/or pick up EZH2 mutations.

In sum, we would argue that the prevalences of GCB-DLBCL among transformed FL (80%) and of type B among diagnostic FL (18%) are not so high that they are incompatible with one another. Further, our published study examining clonal dynamics in progressed vs. transformed FL has shown that the overt genetic signature of FL looks very different from that of subsequently transformed disease (doi.org/10.1371/journal.pmed.1002197), and thus it may be misleading to draw rigid connections between pre- and post-transformation tumor profiles. We have now included an additional paragraph to the discussion to address this point. Also, suggestions of histogenic origin from the original Nature Cancer submission were not included in the version submitted to Nature Comms.

-Use of anthracycline-based chemo for tFL does not to my knowledge eliminate the poor prognostic impact of transformation compared to non-transformed cases. To what extent might selection bias have impacted survival outcomes in this set of patients? (i.e. are cases with

available viable samples different from the general FL population?) This comparison could be performed I believe with data available to these authors.

>> We have now assembled characteristics of the general FL patient population seen at BCCA over the same time period (n=992) and compared these to our CyTOF cohort (**Table S1**). This revealed that indeed our CyTOF cohort is relatively enriched for younger patients with larger tumor masses and who were more likely to have received primary systemic therapy. These features would fit with the fact that we required ~3 million live cells per sample for CyTOF assay and therefore selected for patients with sufficiently large biopsies. Accordingly, our observations will need to be validated in a more representative cohort of FL patients. We have now amended the text to acknowledge this important limitation.

-Is there any evidence for an enrichment in Tfh cells in tFL cases?

>> Among 152 FL cases assessable for transformation, 72 had available T-cell data. We did not observe any significant differences in total Tfh cell content (as a percentage of total T cells) when comparing diagnostic samples that subsequently transformed (n=6) vs. those that did not (n=66) (**Fig R1**). Of note, the T39 subset of Tfh cells trended (p=0.078), but in the opposite direction (more T39 Tfh cells in the not-transformed cases).

Figure R1. Proportion of Tfh cells among total T-cells in diagnostic FL samples that did or did not subsequently transform. NO, did not transform; YES, did transform.

-I was asked also to consider prior review before transfer of the ms to Nat Comms. Some points overlap with above:

-I agree with that reviewer that conclusions regarding implications of this study should be altered. Functional studies and validation in additional cohorts for now are the only future directions with no current avenue for clinical translation therapeutically or for establishment of clinical biomarker assays.

>> We have now amended the text to reflect that the findings presented will need to be validated in independent cohorts prior to any consideration for development as a clinical assay.

-Detailed clinical comparison of these cases to overall FL (as above) can help address some questions of robustness.

>> We have now provided a comparison of cohort characteristics to the general FL patient population as described above.

Minor points:

-How do the authors justify 50% of cells in A or B clusters to assign that case to that subtype? Was this chosen arbitrarily or based on statistical calculations that compared heterogeneity of FL populations across tumors?

>> The 50% cutoff was selected such that every case could be assigned to one and only one MC type. This rule is perhaps better described as assigning each sample to its most abundantly populated MC group; however, the end result is the same as the most abundant MC group in each case included at least 50% of cells (**Fig R2**). This begs the question as to how many samples had a “close second” population that might offer a competing MC designation. In fact, inspection of the 1st and 2nd most populated MC groups within each sample revealed 136/154 (88%) were composed of at least 80% cells of the assigned MC type. There was one sample that stood out, however, with 51.5% MC-C and 48.5% MC-A cells.

Figure R2. Proportion of abnormal B-cells in the 1st and 2nd most abundantly populated MC clusters within each sample. The sample highlighted in red was composed of 51.5% MC-C and 48.5% MC-A cells.

This particular “borderline” case raises the issue as to how robust the described A vs. B vs. nonA/nonB distinctions would be if the MC-A/B content thresholds were lowered. To test this, we examined risk of transformation at reduced MC-A/B content thresholds of 40, 30, 20, and 10%. Since cells of types MC-A and MC-B tend not to co-occur in the same sample, this procedure did not yield any samples fulfilling criteria for both type A and type B tumors.

Notably, differences in transformation risk across tumor types A/B/nonAnonB remained significant at all tested levels with type B always showing the highest risk (**Fig R3**).

Figure R3. Kaplan-Meier curves for risk of transformation using decreasing thresholds for MC-A/B cell content for assigning tumors to types A/B, respectively. Log-rank p-values are shown at each threshold.

Lowering of MC-A/B content thresholds also retained significance in terms of mutational associations down to 30% for A/notA and B/notB comparisons, while the A/B comparison FDR for MEF2B increased from 0.0931 to 0.1061 (**Table R1**). It should be noted here that a heterozygous mutation in a tumor with 50% B-cell content would yield a VAF of 7.5% at the 30% threshold, which approaches the default minimum VAF of 5% for declaring a mutation to be present in a given sample. For this reason, we feel it becomes problematic to consider lower MC thresholds for purposes of mutational associations.

It was not immediately apparent to us what the most suitable statistical approach might be in defining a cutoff point. One approach we looked at was to perform K-means clustering to divide the distribution of all MC abundances across all tumors into 2 groups. This defined one cluster with maximum value 37.5 and the other with minimum value 48.5, suggesting a “natural” segregation point may lie between those two values (**Fig R4**). The only case that would be reclassified by this alternate cutoff would be the one “borderline” MC-C/A (51.5/48.5%) case referred to above, but whose reclassification from type C to A would affect neither transformation risk nor mutation associations as outlined above.

Figure R4. Cell content per tumor for each MC cell type. Each X-axis category includes 154 data points, one for each FL sample. The grey dashed line indicates the 50% threshold. The green dotted lines indicate the extents of K-means clustering defined groups.

In sum, we appreciate that the 50% cutoff value would appear rather arbitrary; however, alternate cutoffs as described above had little to no effect on main conclusions relating to transformation risk and mutational associations. Nonetheless, we have amended the text to indicate that tumor MC types are defined by the most abundantly populated MC group rather than the 50% cutoff value in hopes this will better reflect the priority of assigning each tumor to a single MC type.

Reviewer #5 (Remarks to the Author): expertise in single cell analysis in lymphoma

Wang et. al. present single-cell phenotyping of a large Follicular Lymphoma cohort. They identify a subset of FL tumors with germinal center B cell phenotype associated with an increased risk of transformation. Given that a subset of FL patients progresses to aggressive disease over time, this is a highly relevant study for the field. The cohort size and phenotyping based on multiple assays represents a unique dataset of great value. My concerns regarding the single-cell analysis are as follows:

Major:

1. This is a very difficult story to follow given its current structure. One of the factors that causes the most confusion is switching between phenograph clusters (PGs) and meta clusters (MCs) when describing the B cell results.

>> Thank you for this feedback on the presentation of results. We have now gone back and substantially edited the text to bring more focus on the final clustering by MC and retained reference to PG clusters only where necessary (such as to explain how normal and malignant B-cells are delineated from one another in FL samples).

2. Lines 137-143: 54% of samples are non-A, non-B based on the thresholds used for classification. We are provided little information on what their phenotypes are. Figure S10 is dense and difficult to interpret. Markers in table S4 contain CD44, CD24, Igs that are also considered to be characteristic of A,B subtypes.

>> Thank you for pointing this out. We agree the data as presented in the original Fig S10 was confusingly dense. We have removed this figure, and now present the information in a revised **Table S4** to indicate more clearly which markers are expressed by which MC types. In addition, we hope the new MC-focused presentation of results brings out more clearly the callout to **Fig S4** which graphically depicts how the less abundant MC types relate phenotypically to MC types A/B and to one another. We have also amended the results text to emphasize that the nonA/nonB designation does not refer to a discrete “type” per se, but instead represents a mixed bag of the less abundant MC types C, D, E, F, etc. On that point, we have now replaced the nonA/nonB term with NOS for “not otherwise specified”.

We do appreciate that some features of types MC-A and MC-B are also seen in the other MC types; however, this is a reflection of the fact that each MC type is situated on the global phenotypic map with varying proximity to the more “centrally located” MC types A and B (**Fig S4**). We have now amended the text to point out these phenotypic relationships more clearly, and to acknowledge there will be some overlap in the lists of discriminating markers which depend on the particular comparisons being drawn. For example, since types MC-A and MC-C are proximal to one another in the global phenotypic map, the comparisons (MC-A vs. rest) and (MC-C vs. rest) will contain some overlapping features.

In Fig. 7C, nonA-nonB tumors had the worst survival outcome. It seems like a highly prevalent and important category of tumors that should not be ignored.

>> The nonA/nonB designation indeed describes the largest grouping of tumors (54%); however, it is not a distinct entity per se and in fact comprises a mixed bag of several different MC types. We regret that the “nonA/nonB” terminology does not clearly reflect this, and we have accordingly now replaced it in the text and figures with the hopefully more recognizable term “NOS” for not-otherwise-specified. We hope this new terminology draws parallel to the well-known PTCL NOS and DLBCL NOS designations which are widely recognized as diagnostic “wastebaskets” for tumors whose features are not consistent with any of the other known diagnostic subtypes and which exhibit heterogeneous clinical outcomes.

To illustrate the extent of inter-sample heterogeneity present within the nonA/nonB group (now referred to as type NOS in the revision), we plotted all pairwise Euclidean distances between samples within each of the groups A, B, and NOS. For example, among 5 samples there would be 10 unique pairs, and among 10 samples there would be 45 unique pairs. To enable calculation of inter-sample distances, we defined each sample as a single point in 39-dimensional space based on the median expression value for each of the 39 CyTOF markers. As shown in **Fig S9**, inter-sample distances within the A and B groups are similar to one another, and both much lower than the NOS group, supporting that the NOS group is comparatively much more heterogeneous.

This heterogeneity of phenotypes would suggest that the NOS group subsumes multiple distinct biologies, a subset of which presumably exhibit aggressive clinical behavior. Unfortunately, none of the individual MC types within the NOS group (e.g. C, D, E, F, etc) stand out in terms of transformation risk or survival (see **Fig S19**) or mutational associations (**Table R2**; a few show FDR q-values < 0.1 but each involves only 2 identified mutated samples). It is important to recognize that conclusions regarding outcomes among the component MC types within the NOS category are likely underpowered given the limited numbers of each type.

In sum, we do not regard the NOS category as a discrete tumor type, and its nominal prevalence is inflated due to the many different MC types it subsumes. Poorer survival in this group as a whole is certainly intriguing; however, the diversity of phenotypes would suggest a corresponding diversity of biologies. Of note, while NOS cases indeed comprised 9/11 DSS events, these represented just 9/54, or 17% of NOS samples. We have modified the text to point out heterogeneity of the NOS group, and to emphasize that further study will be required to confirm poorer survival in this group and to identify what mechanisms may underlie this finding.

3. It seems like a missed opportunity that only a superficial analysis from scRNA-seq is presented. The granularity of this assay has the potential to clearly delineate (i) unbiased biology of what distinguishes type A and B tumors,

>> Thank you for this comment. We have now extended our analysis of scRNA-seq data and also added available bulk RNA-seq data. The scRNA-seq data suggested that B-cells from type A and B samples may represent different stages of the GC reaction as evidenced by antigen presentation and heat shock/stress response genes in type B cells (= early stage) vs. ribosomal and immune response genes in type A cells (= late stage). In contrast, the bulk RNA-seq data suggested that local microenvironmental interactions differ with type A samples enriched for ECM remodeling genes vs. type B samples enriched for chemokine signaling genes. These findings are now presented in the new **Figs S12-13** and **Table S6**.

(ii) extent of intrasample heterogeneity including non-A, non-B features.

>> As the goal of the scRNA-seq experiment was to confirm the A vs. B distinction, we intentionally selected relatively pure examples of type A and B tumors from the CyTOF cohort. Nonetheless, single cell resolved data does afford the opportunity to examine intratumoral heterogeneity. Among the 6 selected FL samples, 4 contained small populations (up to 3.5%) of nonA/nonB cells as assessed by CyTOF, while the remaining 2 were either 100% MC-A or MC-B. Within each of the 4 samples containing nonA/nonB cells, there were 4-6 Phenograph clusters identified (**Fig R5**). We examined kappa:lambda mRNA ratios for each of the clusters, and found that they readily parsed as either polytypic (i.e. normal) and monotypic (i.e. malignant). Small populations of polytypic cells in each of the samples (assigned as Naive or Memory clusters) are taken to represent residual normal B-cells. While the dominant monotypic cell populations aligned consistently with either type A (scB03/04/06/08) or type B (scB07/09) clusters, small populations of monotypic cells could also be discerned which were assigned to Doublet, Cycling, or "Other" cell clusters.

Figure R5. PG clustering of scRNA-seq data. UMAP projections of the 4 FL samples with nonA/nonB cells as detected by CyTOF are shown. PG clusters were manually annotated with cell type labels as in **Fig 2B**. Each PG cluster is also annotated with the number of Kappa:Lambda RNA counts, followed by the total number of cells in the cluster in parentheses. “Mono”-typic annotations were assigned for K:L ratios >7 or <0.3; rest were designated as “poly”-typic.

We favor interpreting the Doublet and Cycling clusters as true to their labels; that is, bona fide cell doublets (present in all 4 samples) and cycling cells (present in 2 samples with non-overlapping nonA/nonB populations per CyTOF). This leaves just one sample (eBF167660) with a single cluster of 15 monotypic “Other” cells representing ~1% of total monotypic B-cells in the sample. This does indeed roughly approximate the proportion of nonA/nonB (MC-E) cells in that sample as measured by CyTOF.

In sum, the samples selected for scRNA-seq were relatively pure examples of type A and type B tumors, and thus suboptimal for assessing intratumoral heterogeneity and nonA/nonB cell features in particular. Our analysis above did reveal a potential scRNA-seq correlate (15 “Other”

cells) to a nonA/nonB cell population (MC-E per CyTOF); however, this data is too sparse to support any meaningful interpretation as to the transcriptomic character of nonA/nonB cells.

Which samples were used for scRNA-seq analysis? From Fig. S14, it seems like they were independent from the CyTOF samples? In which case: how were these samples classified into A/B?

>> We apologize that this information was not clearly indicated. The 6 samples used for scRNA-seq were indeed selected from the cohort of CyTOF samples. We have amended Fig S14 (now Fig S10) to indicate more clearly which data types were generated from each of the samples.

If they overlapped, what % of their cells were found to be A/B/nonA-B based on CyTOF analysis?

>> Based on CyTOF analysis, the MC composition of the 6 samples used for scRNA-seq is provided in the screen shot below of relevant rows from Table S5. From this table (and as noted in the response above including Fig R5), the nonA/nonB cell content was less than 3.5% in all cases.

	A	B	C	D	E	F	G	H	I	J	
1	Table S5. FL samples (Composition by B-cell MC cluster)										
2	Indicated values are % of malignant B-cells										
3											
4	Sample Code	2(MC-A)	3(MC-B)	5(MC-C)	7(MC-D)	6(MC-E)	8(MC-F)	4(MC-Mem)	1(MC-Nav)	11(MC-H)	1
5	eBF157798	100	0	0	0	0	0	0	0	0	0
20	eBF156080	98.69138495	0	1.308615049	0	0	0	0	0	0	0
32	eBF157881	98.18423384	0	1.815766165	0	0	0	0	0	0	0
35	eBF146082	96.51878582	0	0	0	0	0	3.48121418	0	0	0
51	eBF146039	0	100	0	0	0	0	0	0	0	0
54	eBF167660	0	98.49315068	0	0	1.506849315	0	0	0	0	0
159											

A wealth of information could be gained from an unbiased analysis of these results rather than using them to validate markers from CyTOF.

>> Thank you again for this suggestion. Please see our response above including new Figs S12-13 and Table S6. The manuscript text has also been amended to incorporate these additional analyses.

4. Fig. 6A mutations seem to overlap in A,B, non A-B. However, Fig. 6B is presented as A vs rest, B vs rest. This makes the results seem more mutually exclusive than they are.

>> Thank you for pointing this out. We have now included an A vs. B plot in Fig 6B (and provided associated statistical comparisons in Table S7) to better illustrate the relationship of mutations across the 3 groups.

Minor:

1. Conclusions are sometimes over-interpreted. Example Line 174: ...“suggesting that phenotypic variation within individual tumors tends to occur in a progressive, incremental manner.”

>> Thank you for this feedback. We have now removed this statement from the text.

2. Perhaps Fig. S14 should appear much earlier in the paper to give us an idea of what assays were performed and on which samples.

>> We have now called out a modified version of this figure earlier in the manuscript text (now **Fig S10**) so it is clear exactly what data is available for each of the samples.

REVIEWERS' COMMENTS

Reviewer #4 (Remarks to the Author):

My concerns have been adequately addressed. In particular comparison of these cases to a larger FL cohort to reveal differences in clinical factors places these data in better context.

Reviewer #5 (Remarks to the Author):

My concerns have been adequately addressed by the authors.

Reviewer #4 (Remarks to the Author):

My concerns have been adequately addressed. In particular comparison of these cases to a larger FL cohort to reveal differences in clinical factors places these data in better context.

Reviewer #5 (Remarks to the Author):

My concerns have been adequately addressed by the authors.

>> All reviewer concerns have been addressed.